# Linear Transformers as VAR Models: Aligning Autoregressive Attention Mechanisms with Autoregressive Forecasting

**Jiecheng Lu** [1]  **Shihao Yang** [1]

## Abstract

Autoregressive attention-based time series forecasting (TSF) has drawn increasing interest, with mechanisms like linear attention sometimes outperforming vanilla attention. However, deeper Transformer architectures frequently misalign with autoregressive objectives, obscuring the underlying vector autoregressive (VAR) structure embedded within linear attention and hindering their ability to capture the data generative processes in TSF. In this work, we first show that a single linear attention layer can be interpreted as a dynamic VAR structure. We then explain that existing multi-layer Transformers have structural mismatches with the autoregressive forecasting objective, which impair interpretability and generalization ability. To address this, we show that by rearranging the MLP, attention, and input-output flow, multi-layer linear attention can also be aligned as a VAR model. Then, we propose Structural Aligned Mixture of VAR (SAMoVAR), a linear Transformer variant that integrates interpretable dynamic VAR weights for multivariate TSF. By aligning the Transformer architecture with autoregressive objectives, SAMoVAR delivers improved performance, interpretability, and computational efficiency, comparing to SOTA TSF models. The code implementation is available at this link.

## 1. Introduction

In recent years, autoregressive decoder-only Transformers have made significant strides (Vaswani, 2017; Radford, 2018), powering Large Language Models (LLMs) (Brown et al., 2020; Touvron et al., 2023) capable of handling complex sequential data. Their core mechanism, autoregressive

self-attention, computes attention weights between the current token and all preceding tokens during prediction. However, using softmax to compute the $N \times N$ attention map results in large $O(N^2)$ time complexity as the sequence length grows. To address the efficiency bottleneck, researchers have developed efficient variants like Linear Transformers (Katharopoulos et al., 2020; Hua et al., 2022). By replacing softmax with a linearizable kernel, linear attention reduces complexity from $O(N^2)$ to $O(N)$ by maintaining a $2d$-dimensional hidden state instead of forming the full $N \times N$ attention map (Choromanski et al., 2021; Hua et al., 2022; Sun et al., 2023). Although it often underperforms vanilla attention in complex tasks like NLP, studies show that in simpler tasks, such as time series forecasting (TSF), linear attention can outperform vanilla attention (Patro & Agneeswaran, 2024; Lu et al., 2024a; Behrouz et al., 2024).

Autoregressive modeling has a long history in time series forecasting. Traditional methods like ARIMA handle univariate series through autoregression, differencing, and moving averages (Winters, 1960; Holt, 2004), while Vector Autoregression (VAR) (Stock & Watson, 2001; Zivot & Wang, 2006) extends this to multivariate settings by capturing cross-variable lag dependencies. Although widely used in fields like economics and climate due to their interpretability and theoretical guarantees (Burbidge & Harrison, 1984; Pretis, 2020), these models' linear assumptions and fixed lag orders limit their ability to capture complex patterns. With growing data scale and complexity, deep learning-based TSF models, especially attention-based approaches, have outperformed traditional AR/VAR methods (Li et al., 2019; Zhou et al., 2021; Nie et al., 2022; Liu et al., 2024).

Previous research offers various perspectives on linear attention, viewing it as an RNN with linear state updates, a dynamic temporal projection, or fast weight programming (Katharopoulos et al., 2020; Schlag et al., 2021). In this paper, we show that linear attention naturally contains a VAR structure. While a single-layer linear attention module can be directly interpreted as a VAR model, stacking multiple layers introduces structural mismatches with the time series generative process, reducing its effectiveness for TSF. We demonstrate that by reorganizing the input-output flow, multi-layer linear attention can fully align with a VAR

[1]Georgia Institute of Technology. Correspondence to: Shihao Yang <shihao.yang@isye.gatech.edu>.

*Proceedings of the 42nd International Conference on Machine Learning*, Vancouver, Canada. PMLR 267, 2025. Copyright 2025 by the author(s).

model. Further, we propose Structural Aligned Mixture of VAR (SAMoVAR), which enhances linear Transformers for TSF, improving both accuracy and interpretability.

The main contributions are summarized as follows:

1) We provide a new perspective by interpreting a single-layer linear attention module as a VAR structure, where the *key* represents the observation and the outer product of value and query forms dynamic VAR weights.

2) We analyze how the designs of existing Transformers lead to misalignments with a VAR model's time series generative objective, including mismatched losses, inconsistent residual streams, and unbalanced observation weighting.

3) We show that properly arranging the input-output flow in a linear Transformer allows multi-layer linear attention to act as a expressive dynamic VAR model. With $l$ layers, each past step's influence on future steps is captured through a "temporal influence path" involving up to $l-1$ intermediate nodes, enhancing interpretability.

4) Based on this aligned structure, we propose SAMoVAR for TSF. Experiments demonstrate that it surpasses previous TSF models in accuracy, interpretability, and efficiency.

## 2. Background

### 2.1. Time Series Forecasting

Time Series Forecasting (TSF) aims to predict future values in a multivariate sequence $\mathbf{S} \in \mathbb{R}^{L \times C}$, split into a historical part $\mathbf{S}_I \in \mathbb{R}^{L_I \times C}$ and a future part $\mathbf{S}_P \in \mathbb{R}^{L_P \times C}$, where $L = L_I + L_P$ are the series lengths, and $C$ is the number of channels. The task is to learn a function $f : \mathbb{R}^{L_I \times C} \to \mathbb{R}^{L_P \times C}$ that generates $\widehat{\mathbf{S}}_P = f(\mathbf{S}_I)$, given the input $\mathbf{S}_I$.

### 2.2. Preliminaries: Attention Mechanisms

Previous studies have examined autoregressive attention mechanisms from various angles, emphasizing their common feature of dynamic weights (Katharopoulos et al., 2020; Hua et al., 2022; Sun et al., 2023). For an input sequence of length $N$ and dimension $d$, represented as $\mathbf{X} \in \mathbb{R}^{N \times d}$, with each token denoted by $\boldsymbol{x}_t \in \mathbb{R}^{1 \times d}$, a single-head autoregressive attention layer is formulated as:

$$\begin{aligned} \text{Attn}(\mathbf{X}) &= \sigma\left(\mathbf{M} \odot \left(\mathbf{Q}\mathbf{K}^\top\right)\right)\mathbf{V}\mathbf{W}_o, \\ \text{with } \mathbf{Q}, \mathbf{K}, \mathbf{V} &= \mathbf{X}\mathbf{W}_q, \mathbf{X}\mathbf{W}_k, \mathbf{X}\mathbf{W}_v, \\ \mathbf{X} &:= \mathbf{X} + \text{Attn}(\text{LN}(\mathbf{X})) \end{aligned} \quad (1)$$

Here, $\mathbf{W}_q, \mathbf{W}_k, \mathbf{W}_v, \mathbf{W}_o \in \mathbb{R}^{d \times d}$ are the projection matrices for query, key, value, and output. The causal mask $\mathbf{M} \in \mathbb{R}^{N \times N}$ ensures autoregressive behavior, with $\mathbf{M}_{ij} = 1\{i \geq j\} - \infty \cdot 1\{i < j\}$, allowing only current and past positions. $\text{Attn}(\cdot)$ and $\text{LN}(\cdot)$ denote the attention and layer normalization functions. When $\sigma$ is softmax (ignoring the

$1/d$ scaling), the mechanism becomes vanilla attention. Replacing $\sigma$ with an identity mapping simplifies it to a linear attention with an identity kernel. Adding an MLP layer after the attention layer, as $\mathbf{X} := \mathbf{X} + \text{MLP}(\text{LN}(\mathbf{X}))$, forms a standard autoregressive Transformer block. We will explore the attention from multiple perspectives (P·). An attention layer dynamically computes a temporal mapping weight matrix $\mathbf{Q}\mathbf{K}^\top \in \mathbb{R}^{N \times N}$ for a sequence of length $N$. For each input step $t$, it generates an attention map over all $N$ tokens as dynamic weights. In vanilla attention, these weights are softmax-normalized to sum to 1. In autoregressive attention, a lower-triangular mask $\mathbf{M}$ ensures each step only attends to positions up to $t$. Thus, the attention layer functions as a variable-length dynamic linear layer on the input value sequence $\mathbf{V}$(Vaswani, 2017; Katharopoulos et al., 2020; Yang et al., 2024).

### P2: Recurrent Form and Autoregression

Autoregressive attention can be viewed as a step-by-step generative process with a recurrent formulation. For the input $\boldsymbol{x}_t$ at step $t$, the output $\boldsymbol{o}_t$ is: $\boldsymbol{o}_t = \frac{\sum_{i=1}^t \sigma(\boldsymbol{q}_t, \boldsymbol{k}_i)\boldsymbol{v}_i}{\sum_{i=1}^t \sigma(\boldsymbol{q}_t, \boldsymbol{k}_i)}$, where $\boldsymbol{q}_t, \boldsymbol{k}_t, \boldsymbol{v}_t \in \mathbb{R}^{1 \times d}$ are query, key, and value vectors at step $t$. When $\sigma(\boldsymbol{q}_t, \boldsymbol{k}_i) = \exp(\boldsymbol{q}_t \boldsymbol{k}_i^\top)$, this represents vanilla attention, relying on all previous keys $\boldsymbol{k}_{\{1,...,t\}}$ and values $\boldsymbol{v}_{\{1,...,t\}}$. If $\sigma(\boldsymbol{q}_t, \boldsymbol{k}_i)$ is derived from a kernel feature map $k(\boldsymbol{q}_t, \boldsymbol{k}_i) = \phi(\boldsymbol{q}_t)\phi(\boldsymbol{k}_i)^\top$, the computation is linearized as: $\boldsymbol{o}_t = \frac{\phi(\boldsymbol{q}_t)\sum_{i=1}^t \phi(\boldsymbol{k}_i)^\top \boldsymbol{v}_i}{\phi(\boldsymbol{q}_t)\sum_{i=1}^t \phi(\boldsymbol{k}_i)^\top}$. This avoids the full $N \times N$ attention map by aggregating past information into a hidden state. Ignoring the denominator, the simplified form is: $\boldsymbol{o}_t = \boldsymbol{q}_t \sum_{i=1}^t \boldsymbol{k}_i^\top \boldsymbol{v}_i$. Here, attention acts like an RNN with a 2D hidden state $\boldsymbol{k}_i^\top \boldsymbol{v}_i \in \mathbb{R}^{d \times d}$ and identity state updates. Studies have shown comparable performance without normalization, so we use this simplified form (Zhai et al., 2021; Mao, 2022; Qin et al., 2022; Sun et al., 2023; Yang et al., 2024). In this view, attention is a dynamic autoregressive model with shared weights $w_{t,i} = \sigma(\boldsymbol{q}_t, \boldsymbol{k}_i)$ across all $d$ channels: $\boldsymbol{o}_t = \sum_{i=1}^t w_{t,i}\boldsymbol{v}_i$.

### P3: Fast Weight Programming 
Fast weight programming (FWP) refers to the process of dynamically determining a set of linear predictor weights for each step in the input sequence, i.e., $\mathbf{W}_{\text{FWP},t} = g(\boldsymbol{x}_1, \ldots, \boldsymbol{x}_t) \in \mathbb{R}^{d \times d}$. In linear attention, this is achieved using a summation aggregator to combine past weight information (Schlag et al., 2021): $\mathbf{W}_{\text{FWP},t} = \sum_{i=1}^t \phi(\boldsymbol{k}_i)^\top \boldsymbol{v}_i$, which serves as a dynamic linear predictor for $\boldsymbol{q}_t$. The final output at step $t$ is then $\boldsymbol{o}_t = \boldsymbol{q}_t \mathbf{W}_{\text{FWP},t}$.

### 2.3. Linear Attention as VAR

Recent TSF research shows that linear attention—without softmax—sometimes outperforms vanilla attention (Patro & Agneeswaran, 2024; Lu et al., 2024a; Behrouz et al., 2024). While it can be viewed as an RNN or a form of fast weight

programming (FWP), these perspectives do not directly link to the autocorrelation or generative nature of TSF data. In this section, we demonstrate that linear attention naturally forms a dynamic VAR structure, making it well-suited for modeling TSF data generation.

**P4: Vector Autoregression** A classic Vector Autoregressive model VAR($p$) with lag $p$ uses parameter matrices $\mathbf{A}_j \in \mathbb{R}^{d \times d}$ to model dependencies on $p$ previous time steps:

$$\boldsymbol{y}_t^\top = \mathbf{A}_1 \boldsymbol{y}_{t-1}^\top + \mathbf{A}_2 \boldsymbol{y}_{t-2}^\top + \cdots + \mathbf{A}_p \boldsymbol{y}_{t-p}^\top + \boldsymbol{u}_t^\top,$$

where $\boldsymbol{y}_j \in \mathbb{R}^{1 \times d}$ represents the observations and $\boldsymbol{u}_t \in \mathbb{R}^{1 \times d}$ is the residual. In the RNN and FWP views, the parameter matrices depend directly on $\boldsymbol{k}_i$ and $\boldsymbol{v}_i$ and act on the query $\boldsymbol{q}_t$ rather than sequentially applying to previous steps as in VAR. However, we can reformulate linear attention to reveal its VAR structure. Since $\boldsymbol{q}_t \boldsymbol{k}_i^\top$ is a scalar, rearranging terms gives: $\boldsymbol{o}_t = \sum_{i=1}^{t} \boldsymbol{k}_i \boldsymbol{q}_t^\top \boldsymbol{v}_i$. By transposing this expression, we obtain the VAR form of a single-layer linear attention mechanism:

$$\boldsymbol{o}_t^\top = \sum_{i=1}^{t} \mathbf{A}_{t,i} \boldsymbol{k}_i^\top, \quad \mathbf{A}_{t,i} = \boldsymbol{v}_i^\top \boldsymbol{q}_t \qquad (2)$$

This defines a VAR($t$) structure with dynamic weights, where the observations are $\boldsymbol{k}_i^\top$ and the rank-1 weight matrices $\mathbf{A}_{t,i} = \boldsymbol{v}_i^\top \boldsymbol{q}_t$ are dynamically generated for each step $t$. Unlike RNNs or FWP, where weights propagate across steps, these matrices are independently generated at each time step. Thus, autoregressive linear attention forms its own VAR($\cdot$) structure at each step, as shown in Figure 1(a).

## 3. Aligning the Objective of Autoregressive Attention with Autoregressive Forecasting

In this section, we show that while a **single linear attention layer** naturally exhibits a dynamic VAR structure, the design of current multi-layer Transformers diverges from the VAR training objective. As a result, these models lose the beneficial VAR properties for time series forecasting (TSF).

Time series data naturally show temporal dependence. A classic VAR model captures both autocorrelation and cross-correlation in the data generation process, with weight matrices $\mathbf{A}_j$ decoupling how past values influence future outcomes. Although standard attention and Transformers are effective for modeling complex sequence relationships (e.g., in NLP), their architecture conflicts with VAR's goal of explicitly representing lag-based dependencies. We outline the key sources of this misalignment below.

**VAR Loss and Position Shifting** A VAR model is designed to directly capture the relationship between past observations and future values. Suppose we rewrite Eq. (2) into a strict VAR model form:

$$\boldsymbol{k}_{t+1}^\top = \boldsymbol{o}_t^\top + \boldsymbol{u}_t^\top = \sum_{i=1}^{t} \underbrace{\boldsymbol{v}_i^\top \boldsymbol{q}_t}_{\mathbf{A}_{t,i}} \boldsymbol{k}_i^\top + \boldsymbol{u}_t^\top, \qquad (3)$$

where $\boldsymbol{u}_t \in \mathbb{R}^{1 \times d}$ is the residual not explained by the dynamic VAR system. To align linear attention with a VAR model, minimizing $\boldsymbol{u}_t$ should be part of the training objective. Adding a loss term for each layer's residual $\boldsymbol{k}_t - \boldsymbol{o}_{t-1}$ might partially achieve this but would conflict with the overall Transformer objective.

In a VAR model, the weights are learned to perform forward shift for each input position. With $l$ attention layers, enforcing this VAR loss would result in $l$ shifts, whereas the Transformer's autoregressive objective only requires a single-step shift at the output. As a result, each layer would need to perform only a fractional shift to stay aligned.

**Residual Stream** Next, we focus on how each attention layer operates. In decoder-only Transformers, a common pre-normalization design (Radford, 2018) includes a residual shortcut from input to output, with each block learning the difference between the current and the next step. The attention layers gather information from previous steps, gradually refining this difference and adding it to the shortcut.

If all attention layers are disabled, the model reduces to a local MLP predictor that maps the current input to the next-step output. With patch tokenization, where each token covers $L_P$ time steps, the model effectively becomes an MLP-based TSF predictor mapping $L_P$ input steps to $L_P$ outputs. Adding an attention layer adjusts the input pattern using past tokens to better match the next-step difference. Recent studies of in-context learning (Zhang et al., 2023; Akyürek et al., 2022) suggest that with more layers, the model can dynamically approximate predictors like gradient descent, ridge regression, or shallow MLPs. Thus, attention layers prioritize refining local predictors through context, rather than explicitly modeling step-by-step generation considering raw observations.

**Input and Output** Comparing the attention output in Eq. (1) with the VAR output in Eq. (3), we see that attention outputs must align with the residual shortcut space, while VAR outputs correspond to the key observations $\boldsymbol{k}$. VAR represent the data generation process, so their outputs should not be treated as residuals. However, by adjusting the weight matrix at each step, we can align the dynamic VAR structure of linear attention with the residual-based objective for transitioning observations to the next step. We call this a "key shortcut," which adds an identity matrix to the dynamic

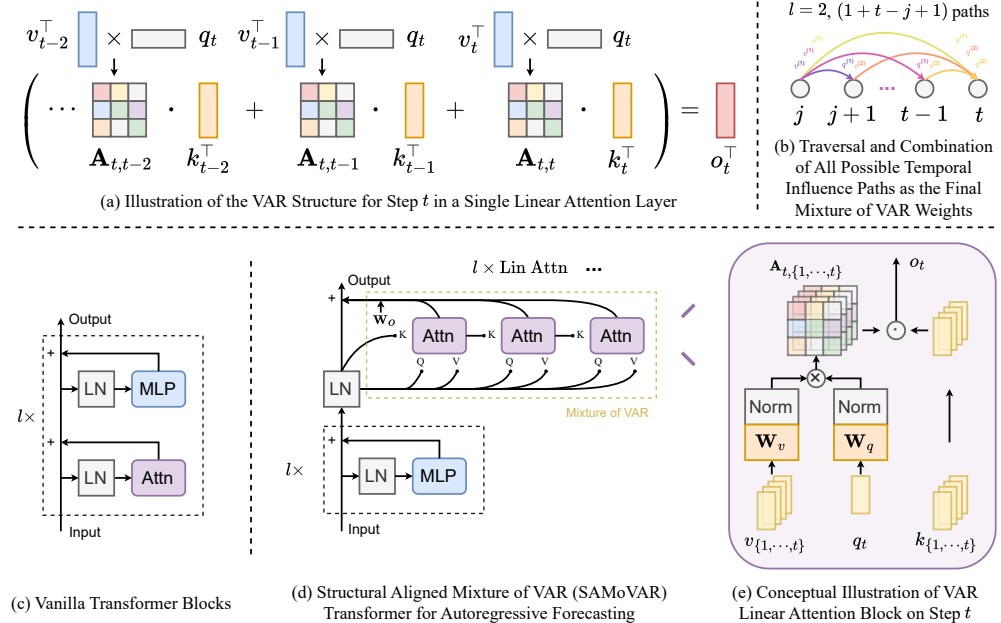

(a) Illustration of the VAR Structure for Step $t$ in a Single Linear Attention Layer

(b) Traversal and Combination of All Possible Temporal Influence Paths as the Final Mixture of VAR Weights

(c) Vanilla Transformer Blocks

(d) Structural Aligned Mixture of VAR (SAMoVAR) Transformer for Autoregressive Forecasting

(e) Conceptual Illustration of VAR Linear Attention Block on Step $t$

*Figure 1.* Visualization of Key Concepts in SAMoVAR. The subfigures highlight different structural and conceptual elements of the model.

VAR weights, guided by an indicator for the output step $t$.

$$\boldsymbol{k}_{t+1}^\top = \boldsymbol{k}_t^\top + \boldsymbol{o}_t^\top + \boldsymbol{u}_t^\top = \sum_{i=1}^{t} \mathbf{A}_{t,i}\,\boldsymbol{k}_i^\top + \boldsymbol{u}_t^\top, \quad (4)$$

$$\text{where} \quad \mathbf{A}_{t,i} = \boldsymbol{v}_i^\top\,\boldsymbol{q}_t + \mathbf{I}\cdot\mathbf{1}_{[i=t]}.$$

However, in a pre-normalization setup, the attention layer only processes the transformed input $\mathrm{LN}(\boldsymbol{x}_t)$ and lacks direct access to the original signal $\boldsymbol{x}_t$ needed to model the residual. As a result, it is not possible to establish a strict VAR recurrence involving $\boldsymbol{o}_t^\top$ and the original signal, regardless of adjustments to $\boldsymbol{k}_t$ or the weight matrices $\mathbf{A}_{t,i}$.

**Balanced Weights of Observations** In a VAR model, all lag positions are initially treated equally, with their influence on future steps determined by learned weights, free from positional bias. In contrast, a multi-layer Transformer requires each token to perform two roles: (1) gather information via attention to predict its next step and (2) serve as context for future tokens. As layers deepen, accumulated residual updates cause the representation $f_{\mathrm{rep}}(x_1, \ldots, x_i)$ to drift away from the original observation semantically. This drift complicates the VAR-like stepwise shift and leads to uneven weighting of original observations in a linear attention-based VAR framework.

# 4. Structural Aligned Mixture of VAR

We show that the misalignments between linear attention and VAR-based forecasting can be resolved by reorganizing the MLP and attention layers in a linear Transformer. By redesigning the input-output flow, we can **enable multi-layer linear attention to maintain a VAR structure**, improving

its ability to model the generative processes of time series data. For a single linear attention layer in Eq. (2), the Transformer naturally aligns with the VAR objective for one-step shifting, as position-wise operations are confined within the layer. The VAR weights remain balanced across past lags when viewed in the original key observation space. To maintain this structure, the output and input signals must follow the same recursive equation, and the residual shortcut should share the same normalization as the attention layer's key inputs, avoiding typical pre-normalization shortcuts seen in standard Transformers.

## 4.1. Multi-layer Linear Attention as VAR

Using a single linear attention layer preserves a clear VAR structure but limits the Transformer's expressive power. Each outer product weight $\boldsymbol{v}_i^\top\boldsymbol{q}_t$ forms a rank-1 matrix, and unlike RNNs or fast weight programming, this VAR formulation cannot increase rank through timestep summation. Below, we show that when multiple linear attention layers are **stacked without MLP layers in between**, and $\boldsymbol{o}_t$ **is directly fed into the next layer**, the attention layers can still function as a **dynamic VAR model**. This model uses the first layer's key input $\boldsymbol{k}_t^{(1)}$ as the observation, with explicit weight matrices that remain aligned with the autoregressive forecasting objective.

Let us denote the output of the first linear attention layer at step $t$ by $\boldsymbol{o}_t^{(1)}$ (omitting residuals, normalization, and setting $\mathbf{W}_o = \mathbf{I}$). In the single-head case: $\boldsymbol{o}_t^{(1)\top} = \sum_{i=1}^{t} \boldsymbol{v}_i^{(1)\top}\boldsymbol{q}_t^{(1)}\boldsymbol{k}_i^{(1)\top}$, where $\boldsymbol{q}_t^{(1)} = \boldsymbol{x}_t\mathbf{W}_q^{(1)}$, $\boldsymbol{k}_i^{(1)} = \boldsymbol{x}_i\mathbf{W}_k^{(1)}$, $\boldsymbol{v}_i^{(1)} = \boldsymbol{x}_i\mathbf{W}_v^{(1)}$. Denote the first layer's key input

by $k_t^{(1)} \equiv k_t$, and let $\mathbf{B}_{t,j}^{(1)}$ be the weight matrix directly acting on $k_j$. Then:

$$o_t^{(1)\top} = \sum_{j=1}^{t} \mathbf{B}_{t,j}^{(1)} k_j^\top, \quad \mathbf{B}_{t,j}^{(1)} = \mathbf{A}_{t,j}^{(1)} = v_j^{(1)\top} q_t^{(1)}.$$

Now take $o_t^{(1)}$ as input to the second layer. The second-layer output becomes: $o_t^{(2)\top} = \sum_{i=1}^{t} \underbrace{(v_i^{(2)\top} q_t^{(2)})}_{\mathbf{A}_{t,i}^{(2)}} k_i^{(2)\top}$, where $q_t^{(2)} = o_t^{(1)} \mathbf{W}_q^{(2)}, \ k_i^{(2)} = o_i^{(1)} \mathbf{W}_k^{(2)}, \ v_i^{(2)} = o_i^{(1)} \mathbf{W}_v^{(2)}$. After expansions and rearrangements, this can be rewritten in a VAR-like form in terms of the original observation $k_t$:

$$o_t^{(2)\top} = \sum_{j=1}^{t} \mathbf{B}_{t,j}^{(2)} k_j^\top, \mathbf{B}_{t,j}^{(2)} = \sum_{i=j}^{t} \underbrace{(v_i^{(2)\top} q_t^{(2)})}_{\mathbf{A}_{t,i}^{(2)}} \mathbf{W}_k^{(2)\top} \mathbf{B}_{i,j}^{(1)}$$

Thus, for $l$ stacked linear attention layers, the final weight $\mathbf{B}_{t,j}^{(l)}$ for the original key observation $k_j$ at step $t$ is:

$$\mathbf{B}_{t,j}^{(l)} = \sum_{i=j}^{t} \underbrace{(v_i^{(l)\top} q_t^{(l)})}_{\mathbf{A}_{t,i}^{(l)}} \mathbf{W}_k^{(l)\top} \mathbf{B}_{i,j}^{(l-1)}. \quad (5)$$

This shows how stacking multiple linear attention layers—without MLP layers in between—produces a expressive dynamic VAR structure for autoregressive forecasting, using the first layer's key input as the observation.

**Temporal Influence Path** The dynamic weight matrix at step $j$ for step $t$, $\mathbf{B}_{t,j}^{(l)}$, is derived by applying the layer-$l$ weights $\mathbf{A}_{t,i}^{(l)}$ and $\mathbf{W}_k^{(l)\top}$ to the previous layer's $\mathbf{B}_{t,j}^{(l-1)}$ across all intermediate steps $i$. Repeated multiplication of different $\mathbf{W}_k$ matrices can lead to numerical instability. To address this, we replace the key projection with the identity matrix $\mathbf{I}$. Under this setup, each term in the summation can be interpreted as a modification (amplification or attenuation) of the previous layer's dynamic weights across the intermediate points after step $j$ to $t$. The iterative factor within each path can be expressed as:

$$\mathbf{P}_{t,j,\{i_1,\cdots,i_{l-1}\}}^{(l)} = \mathbf{A}_{t,i_1}^{(l)} \mathbf{A}_{i_1,i_2}^{(l-1)} \cdots \mathbf{A}_{i_{l-1},j}^{(1)}$$
$$= v_{i_1}^{(l)\top} q_t^{(l)} v_{i_2}^{(l-1)\top} q_{i_1}^{(l-1)} \cdots v_j^{(1)\top} q_{i_{l-1}}^{(1)}$$

where $t \geq i_1 \geq i_2 \geq \cdots \geq i_{l-1} \geq j$. This describes a temporal influence path from step $j$ to $t$ involving $l-1$ intermediate timesteps. All possible combinations of intermediate steps form the complete set of influence paths contributing to $\mathbf{B}_{t,j}^{(l)}$. The number of such paths is given by the binomial coefficient $n_{t,j}^{\text{path}} = \binom{(t-j)+(l-1)}{l-1}$.. Because each path representa a rank-1 matrix, the maximum rank

of $\mathbf{B}_{t,j}^{(l)}$ is $n_{t,j,l}^{\text{path}}$. Each path's scale is controlled by a series of dot-product scalars, e.g., $(q_{i_1}^{(l-1)} v_{i_3}^{(l-2)\top})$. Notably, positions farther from the output step $t$ have more influence paths summed, reflecting the dynamic VAR structure's capacity to capture complex long-term dependencies.

**Robust Path Pruning** To maintain numerical stability, we must prevent exploding values in each path's multiplicative chain. Additionally, with distant timesteps, the large number of possible paths can increase weight variance, making it essential for the model to prune unimportant paths. In this setup, any path with a query-key dot product of zero is effectively pruned.

a) Controlling Exploding Values. We use root mean square layer normalization (RMSNorm) on $q$ and $v$ vectors to prevent their norms from growing excessively. The weight matrices $\mathbf{W}_q$ and $\mathbf{W}_v$ are initialized with low variance, ensuring paths start at small scales, especially those with many intermediate points. b) Passive Pruning of Distant Paths. Multi-heads with a sufficiently large dimension $d$ increases the chance that $q_t$ and $v_i$ become orthogonal, resulting in zero dot products that naturally prune unnecessary paths.

Since each layer's output $o_i$ aggregates multiple influence paths, reusing $o_i$ directly to form the next layer's $q$ and $v$ can further complicate control over numerical stability. In the structure above, all the $q$ and $v$ are used only to generate each layer's dynamic VAR weights. Hence, to maintain stability, we choose to compute $q_t^{(l)}$ and $v_i^{(l)}$ directly from the original first-layer input signals. Specifically, let $k_i^{(1)} = x_i^{(1)}$. For the $l$-th attention layer, we parametrize: $q_t^{(l)} = \text{RMSNorm}^{(l)}(x_t^{(1)} \mathbf{W}_q^{(l)}), v_i^{(l)} = \text{RMSNorm}^{(l)}(x_i^{(1)} \mathbf{W}_v^{(l)}), k_i^{(l)} = o_i^{(l-1)} \mathbf{I}$. In this setup, the total weight at each position is a combination of multi-rank path matrices. Gradient updates amplify the ranks of important paths while suppressing less significant ones, implicitly applying a low-rank regularization.

**Key Shortcut and Mixture of VAR** [1] To address instability from initializing all paths with small weights, we introduce a key shortcut by adding $\mathbf{I}$ to $\mathbf{B}_{t,j}^{(l)}$ when $t = j$, as described in Eq. (4). This effectively incorporates a temporal influence path for $l = 0$ at each step. Additionally, instead of relying solely on the final layer's output $o_t^{(l)}$, we aggregate outputs from all attention layers. This creates a mixture of VAR parameters, with the final output expressed as:

$$o_t^{(\text{final})\top} = \sum_{j=1}^{t} \mathbf{C}_{t,j} x_j^{(1)\top}, \ \mathbf{C}_{t,j} = \sum_{m=1}^{l} \mathbf{B}_{t,j}^{(m)} + \mathbf{I} \cdot \mathbf{1}_{[i=t]}.$$

[1]Please note that even though we use the name "Mixture of VAR," its final form remains an integrated and complete dynamic VAR model. The "mixture" aspect is reflected in the traversal and combination of all possible temporal influence paths.

Here, each $\mathbf{B}_{t,j}^{(m)}$ represents the combination of all temporal influence paths from $j$ to $t$ using up to $m-1$ intermediate points. Consequently, $\mathbf{C}_{t,j}$ contains all paths from $j$ to $t$ with up to $l-1$ intermediate points, with a total of $\sum_{m=1}^{l} n_{t,j,m}^{\text{path}} + 1$ paths (as illustrated in Figure 1(b)).

**Structural VAR** A classic VAR model can be generalized into a Structural VAR (SVAR) (Rubio-Ramirez et al., 2010; Primiceri, 2005) by introducing an invertible matrix $\mathbf{D} \in \mathbb{R}^{d \times d}$: $\mathbf{D}\boldsymbol{y}_t^\top = \sum_{i=1}^{p} \mathbf{A}_i \boldsymbol{y}_{t-i}^\top + \boldsymbol{\epsilon}_t^\top$, where $\mathbf{D}$ captures instantaneous relationships among variables but requires extra constraints for proper identification. The structural shocks $\boldsymbol{\epsilon}_t$ are derived by decomposing residuals $\boldsymbol{u}_t$ using $\mathbf{D}$. Since interpreting individual channel weights in the learned VAR observations $\boldsymbol{k}_i$ can be complex, we use this SVAR form mainly to enhance representational capacity and better interpret residuals. The structural form of our multi-layer linear attention VAR is given as:

$$\boldsymbol{x}_{t+1}^{(1)\top} = \boldsymbol{o}_t^{(\text{final})\top} + \boldsymbol{\epsilon}_t^\top = \sum_{j=1}^{t} \mathbf{D}^{-1} \mathbf{C}_{t,j} \boldsymbol{x}_j^{(1)\top} + \boldsymbol{\epsilon}_t^\top .$$

where $\mathbf{D}^{-1}$ can be viewed as a shared output projection $\mathbf{W}_o^\top$ across $l$ attention layers. We parameterize $\mathbf{W}_o^\top$ via a learnable LU factorization: a lower triangular matrix $\mathbf{L}$ (with diagonal fixed to 1) and an upper triangular matrix $\mathbf{U}$ (diagonal activated by softplus) are multiplied to form $\mathbf{D}$, from which we derive its inverse.

### 4.2. SAMoVAR Transformer

The aligned multi-layer linear attention module described above preserves a valid VAR structure, resolving misalignments in standard linear attention related to training objectives, data generation, input/output spaces, and autoregressive forecasting. Building on this, we introduce SAMoVAR (Structural Aligned Mixture of VAR), which reorganizes MLP and linear attention layers, serving as a drop-in replacement for standard linear Transformers in time series forecasting (TSF), as shown in Figures 1 (c) and (d).

A $l$-layer SAMoVAR Transformer includes: a) MLP Component: Consists of $l$ MLP layers that learn optimal representations for VAR observations. After MLP processing, layer normalization ensures VAR outputs align with the transformed input signals. b) SAMoVAR Attention: Comprises $l$ linear attention layers, parameterized with the robust path pruning methods discussed earlier. A unified residual shortcut across all layers stabilizes training. The overall architecture is shown in Figures 1 (d) and (e).

**Patch-based ARX Tokenization** Prior research highlights the importance of preserving univariate dependencies for effective multivariate time series forecasting (TSF) (Zeng et al., 2023; Nie et al., 2022; Lu et al., 2024b). For multivariate inputs $\mathbf{S}_I$, we adopt an autoregressive (ARX) tokeniza-

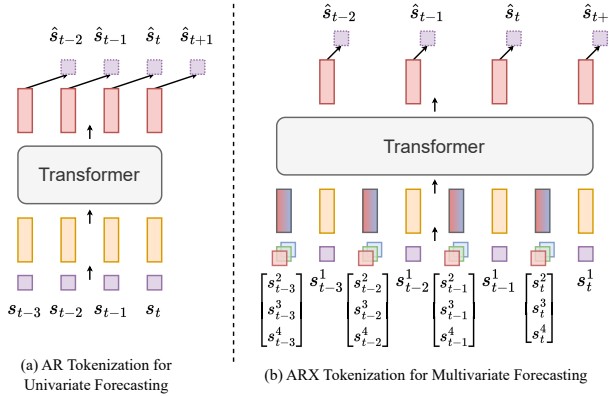

(a) AR Tokenization for Univariate Forecasting

(b) ARX Tokenization for Multivariate Forecasting

*Figure 2.* Illustration of the ARX tokenization, where we use $s_t^j$ to represent the $t$-th patch token of series $j$, $\mathbf{S}_I^{[i:i+L_P, j]}$.

tion strategy, which captures univariate relationships while treating other series as exogenous inputs to model multivariate dependencies. We partition a time series of length $L_I$ into non-overlapping patches of size $L_P$. If needed, zero-padding $P$ is added so that $L_I + P$ is divisible by $L_P$, resulting in $N = \frac{L_I + P}{L_P}$ patches $\mathbf{S}_I^{[i:i+L_P]} \in \mathbb{R}^{L_P \times C}$. For each series $j$, a linear projection $\mathbf{W}_{\text{tok}} \in \mathbb{R}^{d \times L_P}$ transforms its patch $\mathbf{S}_I^{[i:i+L_P, j]}$ into an autoregressive token $(\mathbf{W}_{\text{tok}} \mathbf{S}_I^{[i:i+L_P, j]})^\top \in \mathbb{R}^{1 \times d}$. This token is used to predict the next $L_P$ steps for series $j$, which can be viewed as a PatchTST-style tokenization (Nie et al., 2022) with an autoregressive loss.

Inspired by vector autoregressive with exogenous variables (VARX), $\boldsymbol{y}_t = \sum_{m=1}^{p} \mathbf{A}_m \boldsymbol{y}_{t-m} + \sum_{n=0}^{q} \mathbf{B}_n \boldsymbol{e}_{t-n} + \boldsymbol{u}_t$, where $\boldsymbol{e}_{t-n}$ represents exogenous factors, we model all other series as exogenous when forecasting series $j$. Specifically, a linear projection $W_{\text{ex}} \in \mathbb{R}^{C \times C}$ mixes each channel independently to generate the exogenous token $\left( W_{\text{tok}} \mathbf{S}_I^{[i:i+L_P, :]} W_{\text{ex}}^{[:, j]} \right)^\top \in \mathbb{R}^{1 \times d}$. As shown in Figure 2, we combine autoregressive and exogenous tokens along the sequence dimension to form the input tokens $\mathcal{X}_{\text{input}} \in \mathbb{R}^{C \times 2N \times d}$. The channel dimension is treated as part of the batch for independent computation, with each exogenous token placed before its corresponding target token. Trainable position embeddings based on token positions and channel indices are added. The SAMoVAR Transformer processes the sequence, and the outputs corresponding to target tokens are projected using $\mathbf{W}_{\text{out}} \in \mathbb{R}^{L_P \times d}$ to generate the next-step ARX predictions.

## 5. Experimental Results

### 5.1. Synthetic Tasks

**Model Setup** We use the SAMoVAR architecture described in Fig. 1(d) along with the ARX tokenization from Fig. 2 to train our TSF models. Additionally, we construct a baseline

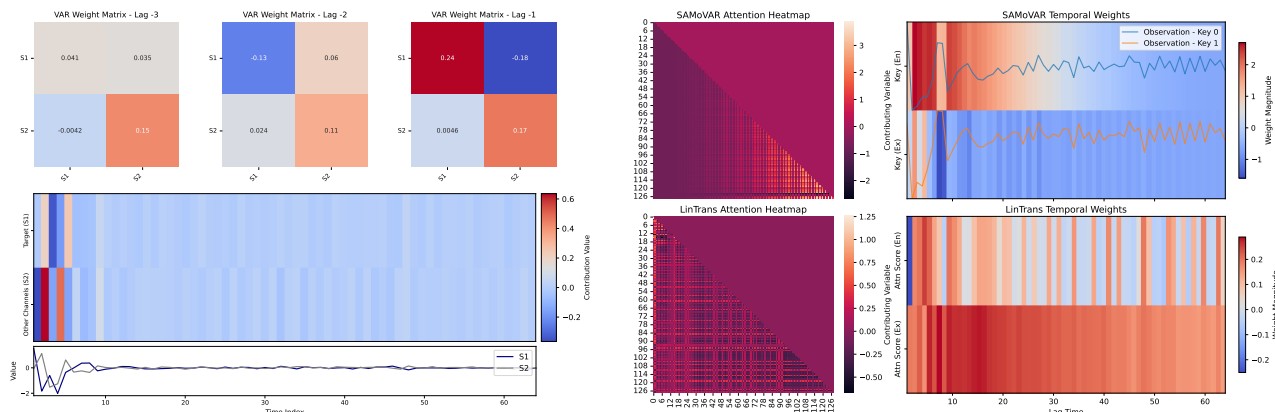

Figure 3. Visualization of the validation datapoint and model weights for the synthetic VAR task. See Section 5.1 for more details.

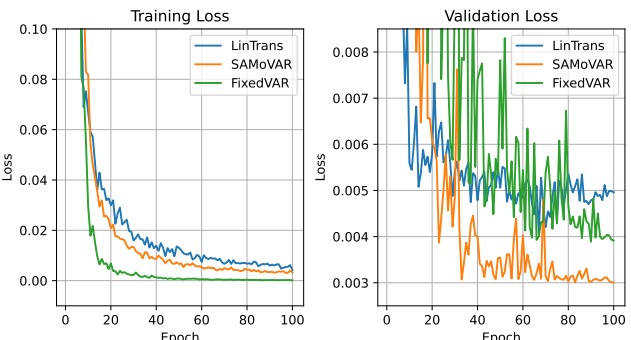

Figure 4. Visualization of the loss curves for synthetic VAR tasks.

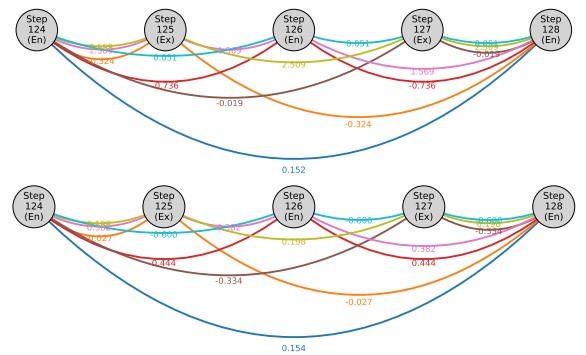

Figure 5. Visualization of the 2 temporal influence paths from step 124 to step 128 for the two series in the datapoint shown in Fig. 3, where even-numbered steps represent endogenous tokens and odd-numbered steps represent exogenous tokens.

model (LinTrans) based on the classic linear Transformer structure shown in Figure 1(c) to highlight the improvements of SAMoVAR. To demonstrate the impact of dynamic VAR weights, we replace SAMoVAR's mixture of VAR modules with a fixed-weight VAR layer (FixedVAR).

**VAR Generalization** To test SAMoVAR's ability to learn and generalize to the underlying data generation process, we generate training and test data using random VAR($p$) models with varying lag orders. For training, $p \in \{1, 2, 3\}$ with coefficients between -0.5 and 0.5. For validation, $p \in \{3, 4, 5\}$

with coefficients between -0.25 and 0.25 to evaluate extrapolation. The input length is $L_I = 64$, output length $L_P = 1$, and $C = 2$ series, with ARX tokenization. We use 3 layers and 2 attention heads. As shown in Fig. 4, during training, FixedVAR effectively memorizes the input structure but struggles in validation. In contrast, SAMoVAR's dynamic VAR inference generalizes well and significantly outperforms FixedVAR. LinTrans, which does not explicitly model the TSF process, performs worse in both phases.

**Explainability** The left side of Fig. 3 shows the original VAR weights, cumulative temporal contributions, and raw observations for a validation datapoint. The middle part compares the averaged attention maps of SAMoVAR and LinTrans. The upper right visualizes SAMoVAR's final output contributions using $\mathbf{W}_{\text{out}} \mathbf{D}^{-1} \mathbf{C}_{t,j} \mathbf{W}_{\text{out}}^{\top}$ across channels. The lower right shows LinTrans's final-row attention map after reordering. SAMoVAR's VAR-based attention better aligns well with the true contribution heatmap, providing more interpretable results than LinTrans.

**Temporal Influence Path** Figure 5 highlights how visualized paths reveal intermediary effects between time steps. The displayed values on the edges represent the averaged weights from the path matrix $\mathbf{P}$. The top section shows Series 1 (S1) and the bottom, Series 2 (S2). From the original VAR weights, we know S2 has a stronger influence on S1 than vice versa. Correspondingly, in the top section, paths passing through exogenous points (marked "Ex") show higher weights, while in the bottom section, these paths are weaker. This visualization effectively uncovers the transmission dynamics within the VAR structure, enhancing interpretability of TSF results.

## 5.2. Multivariate TSF

We conducted comprehensive experiments on 12 widely-used TSF datasets, including Weather, Solar, Electricity

*Table 1.* Summary of Multivariate TSF Results. Averaged test set MSE are reported. See Table 3 for the original results.

| Model | SAMoVAR | LinTrans | FixedVAR | CATS | iTransformer | FITS | PatchTST | Dlinear | EncFormer |
|---|---|---|---|---|---|---|---|---|---|
| Weather | **0.214** | 0.217 | 0.247 | 0.216 | 0.232 | 0.222 | 0.221 | 0.233 | 0.251 |
| Solar | **0.184** | 0.189 | 0.430 | 0.206 | 0.219 | 0.209 | 0.202 | 0.216 | 0.212 |
| ETTh1 | **0.401** | 0.419 | 0.564 | 0.408 | 0.454 | 0.440 | 0.413 | 0.422 | 0.906 |
| ETTh2 | 0.324 | 0.346 | 0.391 | **0.320** | 0.374 | 0.354 | 0.330 | 0.426 | 0.877 |
| ETTm1 | **0.339** | 0.346 | 0.519 | 0.345 | 0.373 | 0.354 | 0.346 | 0.347 | 0.735 |
| ETTm2 | **0.240** | 0.243 | 0.278 | 0.243 | 0.265 | 0.247 | 0.247 | 0.252 | 0.576 |
| ECL | **0.151** | 0.166 | 0.345 | **0.151** | 0.170 | 0.167 | 0.159 | 0.165 | 0.664 |
| Traffic | 0.391 | 0.438 | 0.717 | **0.385** | 0.414 | 0.418 | 0.391 | 0.431 | 0.824 |
| PEMS03 | **0.150** | 0.188 | 0.375 | 0.225 | 0.212 | 0.234 | 0.230 | 0.254 | 0.443 |
| PEMS04 | **0.102** | 0.136 | 0.404 | 0.184 | 0.171 | 0.256 | 0.222 | 0.246 | 0.377 |
| PEMS08 | **0.234** | 0.261 | 0.674 | 0.359 | 0.271 | 0.296 | 0.290 | 0.357 | 0.681 |
| AvgRank | **1.41** | 3.41 | 8.16 | 2.86 | 5.43 | 5.20 | 4.00 | 5.82 | 8.30 |
| #Top1 | **29** | 3 | 0 | 9 | 1 | 0 | 2 | 1 | 0 |

*Table 2.* Summary of Ablation Study Results. Averaged test set MSE are reported. See Table 4, 5, 6, 7 for the original results.

| Exp | SAMoVAR | w/ $\mathbf{W}_k$ | w/o $\mathbf{D}^{-1}$ | w/o QV Norm | Heads=4 dim=16 | Heads=8 dim=8 | Heads=16 dim=4 | Heads=32 dim=2 |
|---|---|---|---|---|---|---|---|---|
| ETTh1 | 0.401 | 0.413 | 0.409 | 0.421 | 0.401 | 0.406 | 0.412 | 0.413 |
| ETTm1 | 0.339 | 0.346 | 0.344 | 0.350 | 0.339 | 0.342 | 0.343 | 0.344 |

| Exp | $l=1$ | $l=2$ | $l=3$ | $l=4$ | $l=5$ | $l=6$ | $l=7$ | $l=8$ |
|---|---|---|---|---|---|---|---|---|
| ETTh1 | 0.420 | 0.411 | 0.401 | 0.404 | 0.413 | 0.414 | 0.408 | 0.410 |
| ETTm1 | 0.346 | 0.342 | 0.339 | 0.345 | 0.346 | 0.348 | 0.349 | 0.351 |

(ECL), ETTs, Traffic, and PEMS [2]. See §A.2 for detailed descriptions of the datasets. Detailed hyperparameter settings and implementation details can be found in §A.4.

For SAMoVAR, LinTrans, and FixedVAR, we used $l=3$ Transformer layers, with the hidden dimension determined empirically as $d = 32\lfloor\sqrt{C}\rfloor$. The number of attention heads was set to $d/16$, ensuring that the dimension per head was 16. During testing, we predicted the next $L_P$ time steps corresponding to the last input token, following the same approach as the baselines for consistency.

**Baselines** In addition to SAMoVAR, LinTrans, and Fixed-VAR, we introduced five recent state-of-the-art baselines: CATS (Lu et al., 2024b), iTransformer (Liu et al., 2024), FITS (Xu et al., 2024), PatchTST (Nie et al., 2022), and DLinear (Zeng et al., 2023). We also included an encoder-only vanilla Transformer, named Encformer, as a comparison to autoregressive Transformers. Encformer uses the same tokenization method as Autoformer (Wu et al., 2021) and Informer (Zhou et al., 2021).

**Fair Comparison** All baseline models were trained under the same conditions with input lengths $L_I \in \{512, 1024, 2048, 4096\}$, and the best performance was reported. This approach may yield stronger results than the original papers, ensuring a rigorous and fair comparison. For the three VAR-based models, we used consistent settings $(L_I : L_P) = (1024, 96), (2048, 192), (2048, 336), (4096, 720)$ to ensure appropriate ARX tokenization.

**Main Results** As shown in Table 1, SAMoVAR consistently outperformed other models across all datasets, with a significantly higher average ranking and top-1 count. Notably,

---

[2]Note: Due to baseline models failing to train on PEMS07 when using batch size = 1 and $L_I = 4096$, this dataset is excluded from the main text, as described in the fair comparison paragraph.

on datasets like Solar and PEMS, which contain many series with stable long-term patterns, SAMoVAR achieved over 30% improvement compared to previous models. This shows the significant benefit of incorporating dynamic VAR structures and alignment when modeling complex TSF data with stable generative processes. Furthermore, although linear Transformers did not fully align with the autoregressive forecasting targets, they still outperformed most baseline models, indicating their potential in TSF.

**Ablation Studies** In Table 2, we present ablation studies to validate the effectiveness of components within SAMoVAR: 1) Reintroducing key projection weights: Based on our analysis in §4.1, introducing $\mathbf{W}_k$ negatively impacts the scaling control of temporal influence paths and increases numerical instability due to the additional weight matrices. The results show a significant performance drop and training instability when $\mathbf{W}_k$ is added. 2) Removing the inverse matrix $\mathbf{D}^{-1}$: This matrix, corresponding to $\mathbf{W}_o$, plays a critical role in controlling the output space. Without it, performance degrades, as expected. 3) Removing RMSNorm from queries and values: As discussed in §4.1, norm control over queries and values is essential for learning effective VAR path weights. Removing RMSNorm significantly degrades performance and causes training issues. 4) Increasing the number of attention heads: When the hidden dimension is fixed, using more attention heads reduces the dimension per head and passively disables more influence paths during initialization, leading to performance degradation. This also reduces the parameter capacity of dynamic VAR weights, further hurting performance. 5) Varying the number of Transformer layers $l$: The layer depth $l$ determines the number of intermediate points in the temporal influence paths included in the final VAR weights. The results show that $l = 1$ yields the worst performance due to the lack of intermediate points. For real-world TSF datasets, $l = 3$ or $l = 4$ (2 or 3 intermediate points) is sufficient for good performance, while larger $l$ values lead to overfitting risks.

**Computational Costs** SAMoVAR introduces no additional computational overhead compared to the vanilla linear Transformer and even reduces the key projection step. This gives it efficiency advantages over other TSF models. Detailed comparisons are shown in Table 8.

## 6. Conclusion and Limitation

This work bridges the gap between linear attention Transformers and VAR models for time series forecasting. We demonstrate that single-layer linear attention inherently captures dynamic VAR structures, while standard multi-layer architectures misalign with autoregressive objectives. By structurally aligning the input-output flow and MLP layers, we propose SAMoVAR, a multi-layer linear Transformer that integrates dynamic VAR weights through temporal in-

fluence paths. SAMoVAR achieves superior accuracy, interpretability, and efficiency compared to state-of-the-art models across synthetic and real-world benchmarks. As for limitation, we have not yet tested larger SAMoVAR models on large-scale general TSF tasks to evaluate their potential as foundation models. Additionally, we have not explored applying SAMoVAR to general sequence modeling tasks to assess whether the learned dynamic VAR weights are effective beyond TSF tasks.

## Impact Statement

This research introduces a novel approach to time series forecasting, advancing the accuracy and interpretability of attention-based time series models. By enabling better-informed decisions in fields such as economics and healthcare, the method demonstrates broad practical value. While its overall societal impact leans toward positive outcomes, the use of this technology in sensitive areas requires thoughtful management and ethical oversight to minimize potential risks and unintended consequences.

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

# A. Appendix

## A.1. Workflow of the SAMoVAR Attention Module

We present the details of the attention module in the SAMoVAR Transformer, as shown in Algorithm 1.

---

**Algorithm 1** SAMoVAR Attention Module

---

**Input:** Sequence $X \in \mathbb{R}^{B \times L \times D}$, layers $L_{\text{attn}}$, head count $H$, head dimension $d = D/H$, query projection $\mathbf{W}_q^{(l)} \in \mathbb{R}^{D \times D}$, value projection $\mathbf{W}_v^{(l)} \in \mathbb{R}^{D \times D}$, invertible matrix $\mathbf{D} \in \mathbb{R}^{H \times d \times d}$, dropout rate $p$
**Output:** Processed sequence $\widetilde{X} \in \mathbb{R}^{B \times L \times D}$
$B, L, D \leftarrow \text{shape}(X)$
$X_{\text{orig}} \leftarrow \text{clone}(X)$ {Store original input for separate Q, V computation}
Initialize $\widetilde{X} \leftarrow X$
Generate invertible matrices $\mathbf{D}$ using LU factorization
**for** $l = 1$ **to** $L_{\text{attn}}$ **do**
  **Compute Query and Value Projections:**
  $Q^{(l)} \leftarrow \text{reshape}(\mathbf{W}_q^{(l)} X_{\text{orig}}, B, L, H, d)$ {$Q^{(l)} \in \mathbb{R}^{B \times L \times H \times d}$}
  $V^{(l)} \leftarrow \text{reshape}(\mathbf{W}_v^{(l)} X_{\text{orig}}, B, L, H, d)$ {$V^{(l)} \in \mathbb{R}^{B \times L \times H \times d}$}
  $Q^{(l)} \leftarrow \text{RMSNorm}(Q^{(l)}), V^{(l)} \leftarrow \text{RMSNorm}(V^{(l)})$
  **Compute Keys from Input:**
  $K \leftarrow \text{reshape}(X, B, L, H, d)$ {$K \in \mathbb{R}^{B \times L \times H \times d}$}
  **Compute Cumulative Fast-Weight Updates:**
  $W \leftarrow \text{cumsum}(K \otimes V^{(l)}, \text{dim} = 1)$ {$W \in \mathbb{R}^{B \times L \times H \times d \times d}$}
  **Compute Output using $Q^{(l)}$:**
  $Y \leftarrow Q^{(l)} \otimes W$ {$Y \in \mathbb{R}^{B \times L \times H \times d}$}
  **Apply Dropout and Mix Output via Structural Matrix D:**
  $Y \leftarrow \text{dropout}(Y, p)$
  $Y_{\text{transformed}} \leftarrow \text{einsum}('blhd, hde-> blhe', Y, \mathbf{D}^{-1})$ {Apply per-head invertible transformation}
  $Y_{\text{transformed}} \leftarrow \text{reshape}(Y_{\text{transformed}}, B, L, D)$ {Reshape back to $\mathbb{R}^{B \times L \times D}$}
  $\widetilde{X} \leftarrow \widetilde{X} + Y_{\text{transformed}}$
  **Update Input for Next Layer:**
  $X \leftarrow Y$
**end for**
**Return:** $\widetilde{X} \in \mathbb{R}^{B \times L \times D}$

---

## A.2. Experimental Datasets

Our multivariate time series forecasting experiments employ twelve real-world benchmark datasets. The original dataset names and their key details are summarized as follows:

**Weather Dataset**[3](Wu et al., 2021) This dataset records 21 meteorological indicators (e.g., temperature, humidity) at 10-minute intervals throughout 2020, collected from the weather station of the Max Planck Institute for Biogeochemistry in Germany.

**Solar Dataset**[4](Lai et al., 2018) Comprising solar power generation data from 137 photovoltaic plants, this dataset captures energy production values sampled every 10 minutes during 2006.

**Electricity Dataset**[5](Wu et al., 2021) Containing hourly electricity consumption records of 321 clients, this dataset spans a three-year period from 2012 to 2014.

**ETT Dataset**[6](Zhou et al., 2021) The Electricity Transformer Temperature (ETT) dataset monitors operational parame-

---

[3] https://www.bgc-jena.mpg.de/wetter/
[4] http://www.nrel.gov/grid/solar-power-data.html
[5] https://archive.ics.uci.edu/ml/datasets/ElectricityLoadDiagrams20112014
[6] https://github.com/zhouhaoyi/ETDataset

ters (including load and oil temperature) from power transformers, recorded at 15-minute (ETTm1/ETTm2) and hourly (ETTh1/ETTh2) resolutions between July 2016 and July 2018. Each subset contains seven critical operational features.

**Traffic Dataset**[7](Wu et al., 2021) This dataset provides hourly road occupancy rates from 862 highway sensors in the San Francisco Bay Area, collected continuously between January 2015 and December 2016.

**PEMS Dataset**[8](Li et al., 2017) A standard benchmark for traffic prediction, the PEMS dataset includes California freeway network statistics recorded at 5-minute intervals. Our experiments utilize four widely adopted subsets: PEMS03, PEMS04, PEMS07, and PEMS08.

### A.3. Related Works

**Time Series Forecasting Models.** Time Series Forecasting (TSF) has seen extensive research spanning traditional statistical models to deep learning-based approaches. Classical methods, including ARIMA (Box et al., 1974) and exponential smoothing (Holt, 2004), effectively model univariate series by capturing trends and seasonality but struggle with complexity in multivariate and nonlinear scenarios. Vector Autoregression (VAR) (Stock & Watson, 2001; Zivot & Wang, 2006) extends autoregressive methods to multivariate series, accounting for cross-variable interactions, and remains prevalent in economics due to interpretability and theoretical robustness (Pretis, 2020).

The emergence of deep learning, particularly RNN-based approaches like LSTM (Hochreiter & Schmidhuber, 1997) and DeepAR (Salinas et al., 2020), significantly improved capturing temporal dependencies. However, RNNs often suffer from challenges in modeling long-range dependencies. Recent advancements leveraging Transformer architectures have transformed TSF through models such as LogTrans (Li et al., 2019), Informer (Zhou et al., 2021), and Autoformer (Wu et al., 2021), each introducing innovative mechanisms like local convolutions, ProbSparse attention, and auto-correlation respectively. Despite their sophistication, these methods have not consistently outperformed simpler approaches like MLPs or linear models in various contexts (Zeng et al., 2023; Xu et al., 2024; Lu et al., 2024b). Recent innovations explore novel directions, such as treating series as patches (Nie et al., 2022), focusing attention across variates (Liu et al., 2024), and utilizing Large Language Models (LLMs) for zero-shot forecasting or foundation model training (Gruver et al., 2023; Jin et al., 2024; Das et al., 2024). Balancing model complexity, interpretability, and effectively modeling both temporal and cross-variate dependencies remains a central challenge in TSF (Zhang & Yan, 2023; Lu et al., 2024b).

**Linear Attention Mechanisms.** The quadratic complexity of traditional Transformer attention (Vaswani, 2017) has spurred extensive research into efficient alternatives. Linear attention mechanisms emerged as a promising solution, beginning with kernel-based linear Transformers (Katharopoulos et al., 2020), which approximate attention operations linearly by employing kernel feature mappings. Despite achieving substantial computational efficiency, initial implementations experienced performance drops compared to vanilla attention.

Subsequent efforts have focused on addressing these shortcomings and enhancing practical usability. Notably, Performers introduced FAVOR+ for unbiased softmax approximation (Choromanski et al., 2021), while TransNormer (Qin et al., 2022) tackled issues of gradient instability and attention dilution. RetNet (Sun et al., 2023) integrated retention mechanisms with linear attention to better capture sequential patterns. Further theoretical insights emerged connecting linear attention to classical concepts like Fast Weight Programming (Mao, 2022; Schlag et al., 2021), providing new interpretability perspectives. Recent advancements have emphasized practical large-scale deployment and efficiency. Gated Linear Attention (Yang et al., 2024) improved hardware efficiency during training, and innovative tiling strategies in Lightning Attention-2 ensured constant training speeds irrespective of sequence length. The introduction of exponential moving-average (EMA) into gated linear attention models further stabilized training dynamics and performance (Ma et al., 2022). Despite these advancements, effectively aligning linear attention structures with generative processes of time series forecasting data remains an active research area.

### A.4. Hyper-parameter Settings and Implementation Details

This section explains the hyper-parameter settings for SAMoVAR attention, linear (AR) attention, and fixed VAR models used in the experiments.

**Attention Module**: We use 3 layers, consisting of 3 MLP layers and 3 attention layers for SAMoVAR attention. For linear

---

[7]http://pems.dot.ca.gov/
[8]http://pems.dot.ca.gov/

attention, we also use 3 Transformer blocks. For fixed VAR, we first use 3 MLP layers and replace the final mixture of VAR modules in SAMoVAR with a single-layer VAR structure with fixed weights. This VAR structure shares the same architecture as linear attention, but the query and key vectors are replaced with a set of fixed position vectors, similar to trainable positional embeddings.

**Hidden Dimension**: For all Transformers, we set the hidden dimension as $d = 32\lfloor\sqrt{C}\rfloor$, where $C$ is the number of multivariate time series. For linear attention and fixed VAR, we use 8 attention heads. For SAMoVAR, the number of heads is determined to ensure each head dimension is 16. For example, for the ETT datasets ($C = 7$), $d = 64$, and the number of heads is 16. For the Weather dataset ($C = 21$), $d = 128$, and the number of heads is 8.

**Initialization**: All linear layers are initialized using a normal distribution with a mean of 0 and a standard deviation of 0.02. Embedding layers are zero-initialized. For projection layers in MLPs, we use GPT-2-style initialization with a scale factor of $\frac{1}{\sqrt{l}}$, where $l$ is the number of MLP/attention layers. The MLP structure follows the standard 2-layer Transformer design with an expansion ratio of 4. The dropout rate is set to 0.1 uniformly.

**Layer Normalization**: We add RMSNorm after query and key projections for SAMoVAR. For the other layer normalization modules, experiments showed no significant difference between traditional layer normalization and RMSNorm, so we opted for RMSNorm due to its lower computational cost.

**Input Preprocessing**: For input time series of shape $(C, L_P)$, where $L_P$ is the input length, we concatenate timestep embeddings along the channel dimension if available (as described in works like Autoformer (Wu et al., 2021) and DLinear (Zeng et al., 2023)). The time series is divided into $N$ non-overlapping patches of size $L_P$. Zero padding is applied if $L_P$ does not divide $L_I$ evenly. The resulting input tokens have shape $(C, N, L_P)$. We apply RevIN (Kim et al., 2022) to normalize each token by subtracting its mean and dividing by the standard deviation of the entire input.

We then create an additional set of tokens by projecting the channel dimension using $C \times C$ linear weights. These exogenous tokens are interleaved with the univariate tokens, resulting in an ARX input of shape $(C, 2N, L_P)$. The patch size dimension $L_P$ is projected to the hidden dimension $d$, resulting in input tokens of shape $(C, 2N, d)$. We add $d$-dimensional channel and position embeddings to the input tokens.

**Transformer Module**: The input tokens are layer-normalized and passed through the Transformer module. The output has shape $(C, 2N, d)$. We select the outputs corresponding to the original univariate tokens, resulting in $(C, N, d)$. This is followed by layer normalization and projection back to dimension $L_P$, producing output tokens of shape $(C, N, L_P)$. The RevIN reverse process restores the outputs by multiplying with the previously calculated standard deviation and adding the mean. We compute the MSE loss between the output and the next timestep's univariate token values.

**Training Setup**: All experiments are conducted on a single Nvidia RTX 4090 GPU with a batch size of 32. For datasets that cause memory overflow, the batch size is reduced to 16 or 8, with 2-step or 4-step **gradient accumulation** to maintain an effective batch size of 32. The optimizer is AdamW with a weight decay of 0.1 and $\beta$ values of (0.9, 0.95). All baseline models are retrained under the same settings.

**Dataset Splits and Preprocessing**: Following Nie et al. (2022); Liu et al. (2024), we use a train-validation-test split ratio of 0.7, 0.1, and 0.2. Input data is standardized using the mean and standard deviation calculated from the training set.

**Training Procedure**: We apply early stopping with a patience of 12 epochs and a maximum of 100 epochs. For the first 5 epochs, we use a warm-up learning rate, gradually increasing it from 0.00006 to 0.0006, followed by linear decay until the maximum epoch is reached.

## A.5. Detailed Experimental Results

## A.6. Additional Visualization of the Synthetic VAR Task

## A.7. Additional Proofs and Clarifications

### A.7.1. COMPLEXITY ANALYSIS OF SAMoVAR ATTENTION

**Proposition.** The computational complexity of SAMoVAR Attention with respect to the sequence length $L$ is $O(L)$.

**Proof.** We provide a detailed complexity analysis for Algorithm 1 (SAMoVAR) here. Let the batch size be $B$, the model dimension $D$, the number of heads $H$, the per-head dimension $d = D/H$, and the number of attention layers $L_{\text{attn}}$.

*Table 3.* Full results of Multivariate TSF task. The test set MSE and MAE are reported. The best results are bolded and the second best are underlined.

| Models | SAMoVAR | | LinTrans | | FixedVAR | | CATS | | iTransformer | | FITS | | PatchTST | | DLinear | | Encformer | |
|---|---|---|---|---|---|---|---|---|---|---|---|---|---|---|---|---|---|---|
| Metric | MSE | MAE | MSE | MAE | MSE | MAE | MSE | MAE | MSE | MAE | MSE | MAE | MSE | MAE | MSE | MAE | MSE | MAE |
| Weather (96) | **0.141** | **0.193** | 0.145 | 0.196 | 0.170 | 0.230 | 0.143 | 0.196 | 0.158 | 0.209 | 0.149 | 0.204 | 0.149 | 0.198 | 0.150 | 0.209 | 0.188 | 0.248 |
| Weather (192) | **0.186** | **0.240** | 0.187 | 0.243 | 0.218 | 0.277 | 0.188 | 0.242 | 0.203 | 0.254 | 0.189 | 0.241 | 0.190 | 0.241 | 0.211 | 0.265 | 0.215 | 0.297 |
| Weather (336) | **0.232** | 0.279 | 0.237 | 0.283 | 0.269 | 0.311 | 0.235 | 0.278 | 0.250 | 0.291 | 0.237 | 0.283 | 0.240 | 0.279 | 0.255 | 0.305 | 0.270 | 0.340 |
| Weather (720) | **0.295** | 0.334 | 0.299 | 0.330 | 0.331 | 0.356 | 0.297 | 0.331 | 0.316 | 0.341 | 0.311 | 0.332 | 0.306 | 0.327 | 0.316 | 0.350 | 0.332 | 0.382 |
| Solar (96) | **0.165** | 0.230 | 0.174 | 0.244 | 0.440 | 0.495 | 0.182 | 0.239 | 0.230 | 0.257 | 0.189 | 0.240 | 0.209 | 0.251 | 0.208 | 0.274 | 0.201 | **0.225** |
| Solar (192) | 0.177 | 0.253 | **0.176** | 0.251 | 0.405 | 0.496 | 0.214 | 0.283 | 0.204 | 0.282 | 0.206 | **0.249** | 0.192 | 0.255 | 0.208 | 0.265 | 0.209 | 0.289 |
| Solar (336) | **0.196** | 0.262 | 0.198 | 0.266 | 0.406 | 0.474 | 0.216 | 0.272 | 0.222 | 0.297 | 0.219 | 0.258 | 0.200 | 0.253 | 0.221 | 0.279 | 0.221 | 0.289 |
| Solar (720) | **0.199** | 0.261 | 0.207 | 0.271 | 0.467 | 0.533 | 0.213 | 0.267 | 0.218 | 0.299 | 0.221 | **0.256** | 0.205 | 0.258 | 0.227 | 0.291 | 0.218 | 0.274 |
| ETTh1 (96) | **0.357** | **0.394** | 0.362 | 0.396 | 0.493 | 0.484 | 0.365 | 0.396 | 0.396 | 0.422 | 0.369 | 0.398 | 0.370 | 0.399 | 0.370 | **0.394** | 0.986 | 0.720 |
| ETTh1 (192) | **0.398** | 0.419 | 0.416 | 0.437 | 0.548 | 0.531 | 0.404 | 0.420 | 0.431 | 0.451 | 0.435 | 0.444 | 0.412 | 0.421 | 0.405 | **0.416** | 0.814 | 0.691 |
| ETTh1 (336) | **0.422** | 0.442 | 0.427 | 0.446 | 0.560 | 0.538 | 0.423 | 0.437 | 0.459 | 0.470 | 0.468 | 0.467 | 0.422 | 0.436 | 0.439 | 0.443 | 0.883 | 0.680 |
| ETTh1 (720) | **0.427** | **0.451** | 0.471 | 0.485 | 0.653 | 0.607 | 0.441 | 0.465 | 0.528 | 0.523 | 0.488 | 0.497 | 0.447 | 0.466 | 0.472 | 0.490 | 0.941 | 0.739 |
| ETTh2 (96) | 0.266 | 0.329 | 0.276 | 0.334 | 0.294 | 0.357 | **0.259** | **0.327** | 0.299 | 0.358 | 0.270 | 0.336 | 0.274 | 0.336 | 0.277 | 0.346 | 1.303 | 0.924 |
| ETTh2 (192) | 0.323 | 0.386 | 0.342 | 0.382 | 0.380 | 0.423 | **0.315** | **0.368** | 0.365 | 0.399 | 0.348 | 0.400 | 0.339 | 0.379 | 0.375 | 0.412 | 0.939 | 0.714 |
| ETTh2 (336) | 0.341 | 0.394 | 0.375 | 0.414 | 0.396 | 0.440 | 0.339 | **0.392** | 0.407 | 0.429 | 0.376 | 0.426 | 0.329 | 0.380 | 0.448 | 0.465 | 0.551 | 0.544 |
| ETTh2 (720) | 0.366 | 0.421 | 0.389 | 0.431 | 0.493 | 0.513 | **0.365** | **0.419** | 0.423 | 0.454 | 0.421 | 0.463 | 0.379 | 0.422 | 0.605 | 0.551 | 0.714 | 0.631 |
| ETTm1 (96) | **0.278** | **0.339** | 0.286 | 0.344 | 0.493 | 0.462 | 0.282 | 0.339 | 0.325 | 0.376 | 0.305 | 0.347 | 0.290 | 0.342 | 0.299 | 0.343 | 0.686 | 0.603 |
| ETTm1 (192) | **0.318** | 0.367 | 0.324 | 0.371 | 0.502 | 0.471 | 0.326 | 0.363 | 0.352 | 0.388 | 0.334 | 0.371 | 0.328 | 0.369 | 0.335 | 0.365 | 0.636 | 0.580 |
| ETTm1 (336) | 0.359 | 0.396 | 0.363 | 0.397 | 0.496 | 0.472 | 0.358 | 0.382 | 0.382 | 0.405 | 0.363 | 0.387 | 0.359 | 0.392 | 0.359 | 0.386 | 0.791 | 0.602 |
| ETTm1 (720) | 0.401 | 0.413 | 0.409 | 0.432 | 0.583 | 0.527 | 0.414 | 0.416 | 0.432 | 0.434 | 0.412 | **0.409** | 0.405 | 0.415 | **0.396** | 0.409 | 0.825 | 0.641 |
| ETTm2 (96) | **0.154** | **0.242** | 0.158 | 0.246 | 0.188 | 0.282 | 0.158 | 0.248 | 0.187 | 0.281 | 0.164 | 0.253 | 0.165 | 0.255 | 0.184 | 0.283 | 0.481 | 0.525 |
| ETTm2 (192) | **0.208** | **0.287** | 0.213 | 0.287 | 0.244 | 0.318 | 0.211 | 0.285 | 0.232 | 0.311 | 0.211 | 0.292 | 0.214 | 0.289 | 0.218 | 0.301 | 0.434 | 0.517 |
| ETTm2 (336) | **0.257** | 0.333 | 0.263 | 0.326 | 0.302 | 0.361 | 0.261 | **0.322** | 0.281 | 0.342 | 0.259 | 0.334 | 0.266 | 0.328 | 0.263 | 0.333 | 0.461 | 0.531 |
| ETTm2 (720) | 0.340 | 0.372 | 0.339 | 0.378 | 0.377 | 0.421 | **0.340** | **0.371** | 0.358 | 0.392 | 0.352 | 0.377 | 0.344 | 0.376 | 0.341 | 0.388 | 0.928 | 0.739 |
| ECL (96) | 0.129 | 0.226 | 0.149 | 0.265 | 0.315 | 0.401 | **0.127** | **0.223** | 0.132 | 0.226 | 0.144 | 0.246 | 0.129 | 0.222 | 0.135 | 0.232 | 0.227 | 0.342 |
| ECL (192) | **0.141** | 0.243 | 0.160 | 0.268 | 0.382 | 0.457 | 0.143 | 0.241 | 0.166 | 0.269 | 0.149 | 0.249 | 0.147 | 0.240 | 0.151 | 0.249 | 0.658 | 0.611 |
| ECL (336) | 0.156 | 0.255 | 0.171 | 0.299 | 0.307 | 0.398 | **0.155** | 0.253 | 0.176 | 0.270 | 0.168 | 0.262 | 0.163 | 0.259 | 0.169 | 0.267 | 0.988 | 0.801 |
| ECL (720) | **0.176** | 0.281 | 0.183 | 0.279 | 0.375 | 0.446 | 0.179 | **0.273** | 0.206 | 0.283 | 0.206 | 0.303 | 0.197 | 0.290 | 0.203 | 0.301 | 0.781 | 0.739 |
| Traffic (96) | 0.371 | 0.261 | 0.423 | 0.294 | 0.764 | 0.487 | 0.359 | 0.253 | 0.356 | 0.259 | 0.404 | 0.286 | 0.360 | 0.249 | 0.399 | 0.286 | 0.915 | 0.460 |
| Traffic (192) | 0.375 | 0.268 | 0.434 | 0.295 | 0.769 | 0.460 | 0.373 | 0.258 | 0.410 | 0.278 | 0.406 | 0.274 | 0.379 | 0.256 | 0.423 | 0.287 | 0.723 | 0.485 |
| Traffic (336) | 0.390 | 0.279 | 0.439 | 0.299 | 0.670 | 0.418 | **0.384** | 0.274 | 0.431 | 0.281 | 0.412 | 0.279 | 0.392 | 0.264 | 0.436 | 0.296 | 0.764 | 0.451 |
| Traffic (720) | 0.429 | 0.298 | 0.454 | 0.306 | 0.663 | 0.431 | **0.425** | 0.298 | 0.458 | 0.299 | 0.449 | 0.291 | 0.432 | 0.286 | 0.466 | 0.315 | 0.892 | 0.483 |
| PEMS03 (96) | **0.119** | 0.232 | 0.144 | 0.258 | 0.299 | 0.437 | 0.157 | 0.267 | 0.135 | **0.229** | 0.193 | 0.274 | 0.198 | 0.285 | 0.198 | 0.299 | 0.162 | 0.272 |
| PEMS03 (192) | 0.148 | **0.248** | 0.146 | 0.264 | 0.418 | 0.523 | 0.197 | 0.300 | 0.198 | 0.299 | 0.221 | 0.299 | 0.201 | 0.288 | 0.231 | 0.328 | 0.471 | 0.513 |
| PEMS03 (336) | **0.191** | 0.279 | 0.217 | 0.315 | 0.344 | 0.431 | 0.222 | 0.318 | 0.234 | 0.330 | 0.238 | 0.313 | 0.218 | 0.304 | 0.254 | 0.351 | 0.542 | 0.548 |
| PEMS03 (720) | **0.142** | 0.248 | 0.245 | 0.306 | 0.440 | 0.509 | 0.325 | 0.406 | 0.279 | 0.344 | 0.285 | 0.353 | 0.304 | 0.392 | 0.331 | 0.413 | 0.597 | 0.610 |
| PEMS04 (96) | **0.092** | **0.193** | 0.117 | 0.216 | 0.268 | 0.383 | 0.141 | 0.253 | 0.121 | 0.217 | 0.215 | 0.299 | 0.221 | 0.320 | 0.209 | 0.301 | 0.116 | 0.232 |
| PEMS04 (192) | **0.100** | **0.198** | 0.132 | 0.250 | 0.583 | 0.604 | 0.189 | 0.310 | 0.165 | 0.279 | 0.238 | 0.331 | 0.195 | 0.294 | 0.228 | 0.323 | 0.436 | 0.482 |
| PEMS04 (336) | **0.107** | **0.207** | 0.139 | 0.256 | 0.341 | 0.434 | 0.196 | 0.309 | 0.175 | 0.290 | 0.265 | 0.355 | 0.226 | 0.318 | 0.252 | 0.343 | 0.432 | 0.483 |
| PEMS04 (720) | **0.109** | **0.209** | 0.157 | 0.271 | 0.422 | 0.468 | 0.209 | 0.327 | 0.221 | 0.331 | 0.306 | 0.391 | 0.246 | 0.344 | 0.294 | 0.377 | 0.523 | 0.538 |
| PEMS08 (96) | **0.138** | 0.213 | 0.168 | 0.249 | 0.977 | 0.774 | 0.212 | 0.276 | 0.170 | **0.225** | 0.337 | 0.322 | 0.183 | 0.236 | 0.318 | 0.326 | 0.253 | 0.302 |
| PEMS08 (192) | **0.197** | **0.224** | 0.247 | 0.284 | 0.569 | 0.503 | 0.289 | 0.313 | 0.269 | 0.295 | 0.343 | 0.361 | 0.273 | 0.305 | 0.325 | 0.344 | 0.703 | 0.619 |
| PEMS08 (336) | **0.298** | **0.313** | 0.306 | 0.328 | 0.565 | 0.471 | 0.324 | 0.338 | 0.303 | 0.324 | 0.352 | 0.349 | 0.335 | 0.332 | 0.374 | 0.362 | 0.831 | 0.685 |
| PEMS08 (720) | 0.304 | **0.250** | 0.321 | 0.340 | 0.583 | 0.515 | 0.359 | 0.369 | 0.342 | 0.351 | 0.403 | 0.387 | 0.369 | 0.376 | 0.411 | 0.415 | 0.936 | 0.716 |

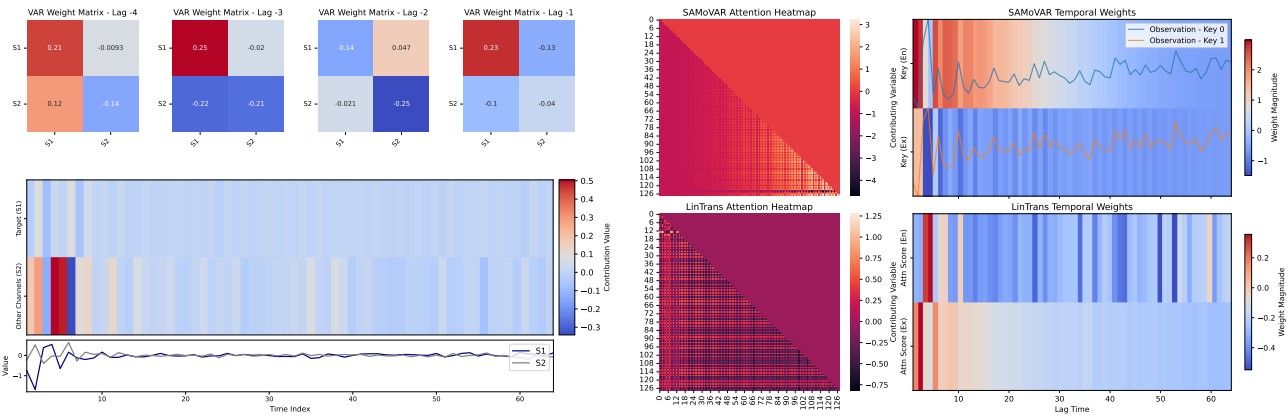

*Figure 6.* Additional Visualization of the VAR Synthetic Task with a Random Datapoint in the Validation Set.

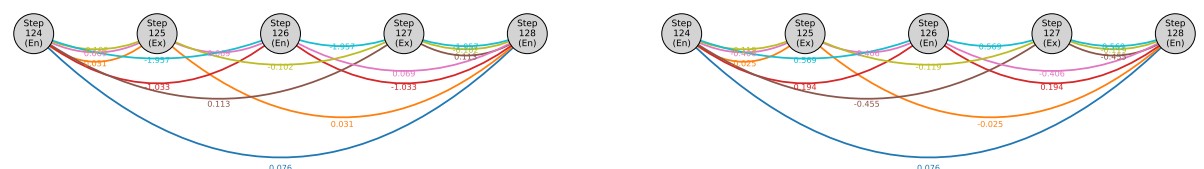

*Figure 7.* Visualization of the 2 temporal influence paths from step 124 to step 128 for the two input time series variable for the datapoint shown above, where even-numbered steps represent endogenous tokens and odd-numbered steps represent exogenous tokens.

**Preprocessing:**

*Table 4.* Full results of the ablation studies of the SAMoVAR module structure. The MSE and MAE of the test set are reported. The best results are bolded and the second best are underlined.

| Models | SAMoVAR | | w/ $\mathbf{W}_k$ | | w/o $\mathbf{D}^{-1}$ | | w/o QV Norm | |
|---|---|---|---|---|---|---|---|---|
| Metric | MSE | MAE | MSE | MAE | MSE | MAE | MSE | MAE |
| ETTh1 (96) | **0.357** | **0.394** | 0.362 | 0.396 | 0.360 | 0.395 | 0.370 | 0.403 |
| ETTh1 (192) | **0.398** | **0.419** | 0.419 | 0.434 | 0.404 | 0.430 | 0.423 | 0.441 |
| ETTh1 (336) | **0.422** | **0.442** | 0.431 | 0.447 | 0.436 | 0.455 | 0.442 | 0.461 |
| ETTh1 (720) | **0.427** | **0.451** | 0.441 | 0.467 | 0.435 | 0.461 | 0.450 | 0.478 |
| ETTm1 (96) | **0.278** | **0.339** | 0.284 | 0.341 | 0.282 | 0.343 | 0.285 | 0.346 |
| ETTm1 (192) | **0.318** | **0.367** | 0.327 | 0.371 | 0.322 | 0.373 | 0.334 | 0.379 |
| ETTm1 (336) | **0.359** | **0.396** | 0.368 | 0.400 | 0.363 | 0.397 | 0.373 | 0.403 |
| ETTm1 (720) | **0.401** | **0.413** | 0.405 | 0.421 | 0.408 | 0.426 | 0.409 | 0.427 |

*Table 5.* Full results of the ablation studies of the number of heads and dimension. The MSE and MAE of the test set are reported. The best results are bolded and the second best are underlined.

| Models | Heads=4,dim=16 | | Heads=4,dim=16 | | Heads=4,dim=16 | | Heads=4,dim=16 | |
|---|---|---|---|---|---|---|---|---|
| Metric | MSE | MAE | MSE | MAE | MSE | MAE | MSE | MAE |
| ETTh1 (96) | **0.357** | **0.394** | **0.357** | 0.395 | 0.363 | 0.398 | 0.365 | 0.401 |
| ETTh1 (192) | **0.398** | **0.419** | 0.403 | 0.424 | 0.415 | 0.426 | 0.418 | 0.431 |
| ETTh1 (336) | **0.422** | **0.442** | 0.436 | 0.449 | 0.435 | 0.449 | 0.439 | 0.452 |
| ETTh1 (720) | **0.427** | **0.451** | 0.429 | 0.452 | 0.433 | 0.454 | 0.431 | 0.453 |
| ETTm1 (96) | **0.278** | **0.339** | 0.281 | 0.343 | 0.280 | 0.341 | 0.283 | 0.346 |
| ETTm1 (192) | **0.318** | **0.367** | 0.322 | 0.369 | 0.322 | 0.371 | 0.323 | 0.373 |
| ETTm1 (336) | **0.359** | **0.396** | 0.362 | 0.399 | 0.364 | 0.401 | 0.365 | 0.401 |
| ETTm1 (720) | **0.401** | **0.413** | 0.404 | **0.413** | 0.406 | 0.414 | 0.406 | 0.414 |

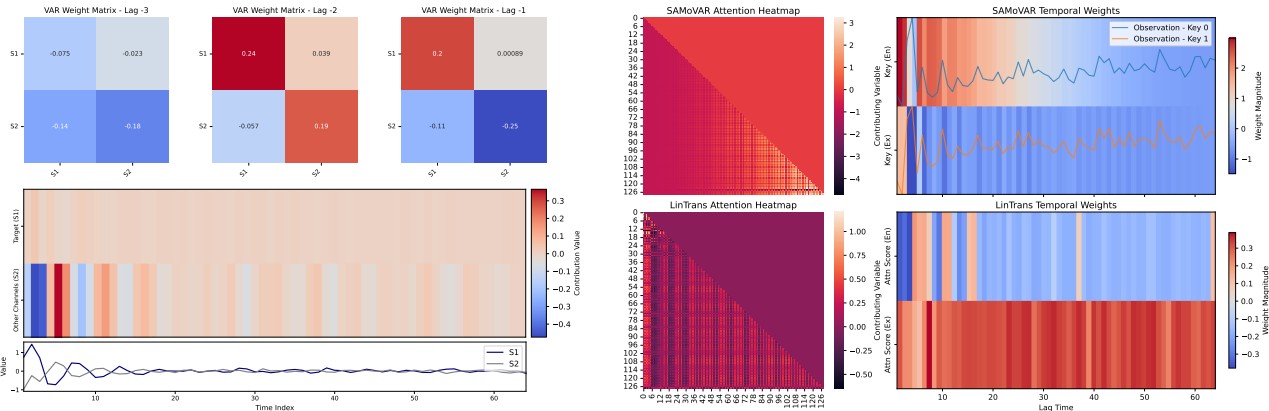

*Figure 8.* Additional Visualization of the VAR Synthetic Task with a Random Datapoint in the Validation Set.

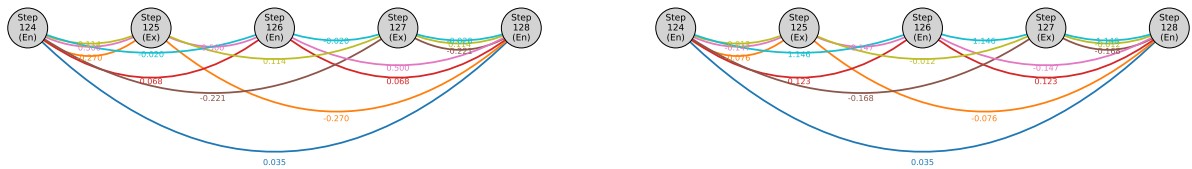

*Figure 9.* Visualization of the 2 temporal influence paths from step 124 to step 128 for the two input time series variable for the datapoint shown above, where even-numbered steps represent endogenous tokens and odd-numbered steps represent exogenous tokens.

- **LU Matrix Generation**: Creating the invertible matrix $\mathbf{D}$ via LU decomposition is independent of sequence length $L$, with complexity $O(Hd^2)$ per layer.

**Per-Layer Operations (for each attention layer $l = 1, \ldots, L_{\mathbf{attn}}$):**

1. **Linear Projections**: Computing query $\mathbf{Q}^{(l)}$ and value $\mathbf{V}^{(l)}$ projections costs $O(LD^2)$.

*Table 6.* Full results of the ablation studies of different number of layers / intermediate points in temporal influence paths. The MSE and MAE of the test set are reported. The best results are bolded and the second best are underlined.

| Models | $l=1$ | | $l=2$ | | $l=3$ | | $l=4$ | | $l=5$ | | $l=6$ | | $l=7$ | | $l=8$ | |
|---|---|---|---|---|---|---|---|---|---|---|---|---|---|---|---|---|
| Metric | MSE | MAE | MSE | MAE | MSE | MAE | MSE | MAE | MSE | MAE | MSE | MAE | MSE | MAE | MSE | MAE |
| ETTh1 (96) | 0.368 | 0.400 | **0.356** | **0.393** | 0.357 | 0.394 | 0.360 | 0.397 | 0.362 | 0.397 | 0.360 | 0.395 | 0.357 | 0.394 | 0.358 | 0.394 |
| ETTh1 (192) | 0.409 | 0.431 | 0.402 | 0.424 | **0.398** | **0.419** | 0.400 | 0.421 | 0.400 | 0.422 | 0.400 | 0.424 | 0.406 | 0.426 | 0.400 | 0.423 |
| ETTh1 (336) | 0.464 | 0.475 | 0.439 | 0.453 | **0.422** | **0.442** | 0.428 | 0.449 | 0.439 | 0.455 | 0.440 | 0.450 | 0.429 | 0.444 | 0.426 | 0.445 |
| ETTh1 (720) | 0.440 | 0.464 | 0.445 | 0.462 | **0.427** | **0.451** | 0.428 | 0.452 | 0.451 | 0.465 | 0.454 | 0.466 | 0.440 | 0.454 | 0.456 | 0.476 |
| ETTm1 (96) | 0.287 | 0.344 | 0.280 | 0.341 | **0.278** | **0.339** | 0.286 | 0.344 | 0.288 | 0.347 | 0.285 | 0.343 | 0.287 | 0.342 | 0.290 | 0.349 |
| ETTm1 (192) | 0.325 | 0.369 | 0.322 | 0.368 | **0.318** | **0.367** | 0.327 | 0.371 | 0.329 | 0.372 | 0.331 | 0.373 | 0.331 | 0.375 | 0.332 | 0.374 |
| ETTm1 (336) | 0.365 | 0.398 | 0.364 | 0.397 | **0.359** | **0.396** | 0.366 | 0.401 | 0.366 | 0.400 | 0.368 | 0.402 | 0.371 | 0.405 | 0.372 | 0.403 |
| ETTm1 (720) | 0.407 | 0.419 | 0.403 | 0.416 | 0.401 | **0.413** | **0.400** | 0.417 | 0.401 | 0.415 | 0.408 | 0.424 | 0.407 | 0.426 | 0.409 | 0.430 |

*Table 7.* Effect of Different Random Seeds on the Results of SAMoVAR. The results show that there is almost no difference across five runs with different seeds, demonstrating the stability of SAMoVAR with respect to random initialization.

| Models | Seed=2023 | | Seed=2024 | | Seed=2025 | | Seed=2026 | | Seed=2027 | |
|---|---|---|---|---|---|---|---|---|---|---|
| Metric | MSE | MAE | MSE | MAE | MSE | MAE | MSE | MAE | MSE | MAE |
| ETTh1 (96) | 0.356 | 0.393 | 0.357 | 0.394 | 0.357 | 0.394 | 0.358 | 0.395 | 0.357 | 0.395 |
| ETTh1 (192) | 0.401 | 0.421 | 0.397 | 0.42 | 0.398 | 0.419 | 0.404 | 0.422 | 0.401 | 0.419 |
| ETTh1 (336) | 0.424 | 0.446 | 0.421 | 0.441 | 0.422 | 0.442 | 0.423 | 0.449 | 0.425 | 0.452 |
| ETTh1 (720) | 0.429 | 0.454 | 0.431 | 0.449 | 0.427 | 0.451 | 0.425 | 0.452 | 0.428 | 0.453 |
| ETTm1 (96) | 0.279 | 0.34 | 0.279 | 0.339 | 0.278 | 0.339 | 0.278 | 0.34 | 0.277 | 0.338 |
| ETTm1 (192) | 0.317 | 0.367 | 0.318 | 0.367 | 0.318 | 0.367 | 0.317 | 0.367 | 0.319 | 0.368 |
| ETTm1 (336) | 0.359 | 0.397 | 0.358 | 0.396 | 0.359 | 0.396 | 0.359 | 0.398 | 0.361 | 0.397 |
| ETTm1 (720) | 0.401 | 0.413 | 0.402 | 0.413 | 0.401 | 0.413 | 0.403 | 0.414 | 0.4 | 0.412 |

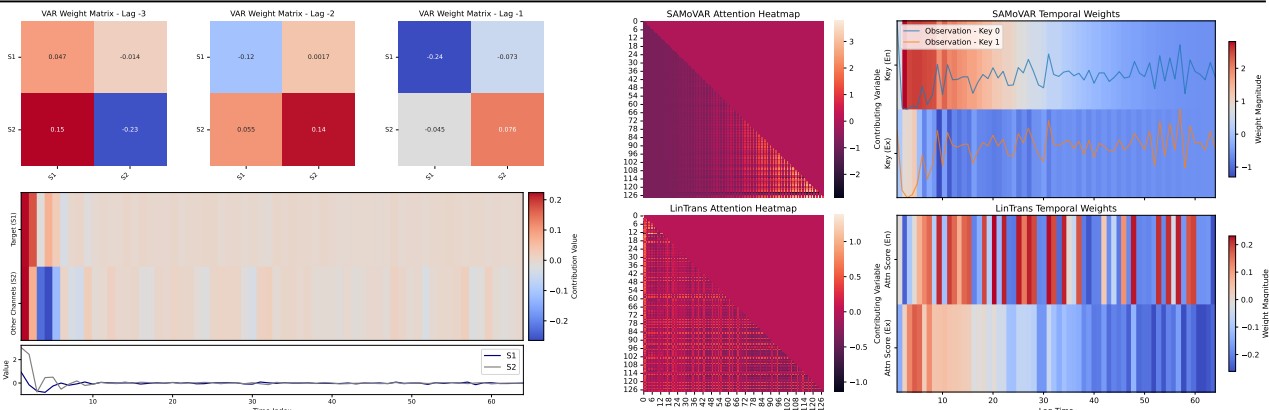

*Figure 10.* Additional Visualization of the VAR Synthetic Task with a Random Datapoint in the Validation Set.

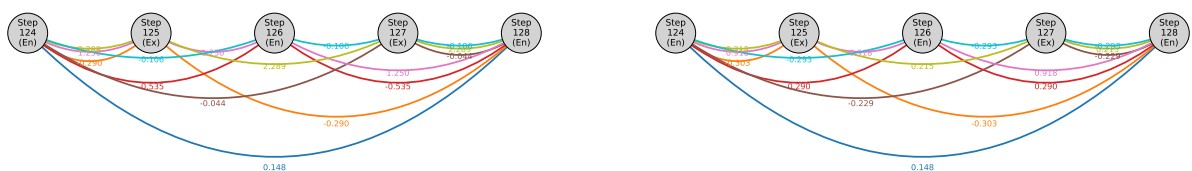

*Figure 11.* Visualization of the 2 temporal influence paths from step 124 to step 128 for the two input time series variable for the datapoint shown above, where even-numbered steps represent endogenous tokens and odd-numbered steps represent exogenous tokens.

2. **Cumulative State Update**: The recursive computation

$$W_t = W_{t-1} + K_t \otimes V_t^{(l)}$$

incurs a complexity of $O(Hd^2)$ per timestep, totaling $O(LHd^2) = O(LD^2/H)$ across all timesteps.

3. **Output Computation**: Calculating

$$Y_t = Q_t^{(l)} \otimes W_t$$

also results in $O(LD^2/H)$.

*Table 8.* Comparison of computational costs. This comparison utilizes the data format of ETTh1 to construct model inputs. $L_I$ is set to 512 for the baselines, 1024 for VAR-based models, and the other hyper-parameters for every model are set according to their default configurations.

| Models | SAMoVAR | | LinTrans | | FixedVAR | |
|---|---|---|---|---|---|---|
| Metric | FLOPs | Params | FLOPs | Params | FLOPs | Params |
| $L_P = 96$ | 43.31M | 157.3K | 50.37M | 181.9K | 35.74M | 196.5K |
| $L_P = 192$ | 25.24M | 175.9K | 29.08M | 200.4K | 21.11M | 215K |
| $L_P = 336$ | 18.44M | 199.6K | 20.99M | 224.1K | 15.69M | 238.7K |
| $L_P = 720$ | 11.38M | 272.6K | 12.63M | 297.2K | 10M | 311.8K |
| Models | Encformer | | PatchTST | | iTransformer | |
| Metric | FLOPs | Params | FLOPs | Params | FLOPs | Params |
| $L_P = 96$ | 1.328G | 1.646M | 180.9M | 1.841M | 42.39M | 1.923M |
| $L_P = 96$ | 1.442G | 1.647M | 215.5M | 3.414M | 42.66M | 1.935M |
| $L_P = 96$ | 1.613G | 1.648M | 267.5M | 5.773M | 43.07M | 1.954M |
| $L_P = 96$ | 2.068G | 1.65M | 405.9M | 12.07M | 44.15M | 2.003M |

*Figure 12.* Additional Visualization of the VAR Synthetic Task with a Random Datapoint in the Training Set.

*Figure 13.* Visualization of the 2 temporal influence paths from step 124 to step 128 for the two input time series variable for the datapoint shown above, where even-numbered steps represent endogenous tokens and odd-numbered steps represent exogenous tokens.

4. **Structural Transformation**:

$$Y_{t,\text{transformed}} = \text{einsum}('bhd, hde \rightarrow bhe', Y_t, \mathbf{D}^{-1})$$

similarly requires $O(LD^2/H)$.

**Overall Complexity:** Summing up each attention layer's operations, each layer requires $O(LD^2)$. Given that the number of attention layers $L_{\text{attn}}$ is constant with respect to sequence length, the overall computational complexity is thus $O(LD^2)$, clearly linear with respect to the sequence length $L$.

Therefore, SAMoVAR Attention achieves linear complexity in sequence length.

A.7.2. CLARIFICATION ON COMPUTATIONAL EFFICIENCY

We further clarify the computational efficiency results presented in Table 8. The values for FLOPs (floating-point operations) and parameter counts were calculated consistently across all compared models using the ETTh1 dataset. Specifically, all baseline models were evaluated using an input length of $L_I = 512$, while linear-attention-based models (including

SAMoVAR) were evaluated using $L_I = 1024$. All other hyperparameters strictly followed the original ETTh1 dataset configurations. For linear attention variants (including SAMoVAR), we set the hidden dimension as $d = \lfloor 32\sqrt{C} \rfloor$, resulting in $d = 64$ for the ETTh1 dataset with $C = 7$. This setting naturally reduces the number of parameters compared to baseline models (PatchTST/iTransformer), which typically use higher dimensions (128/256). SAMoVAR further optimizes computational costs by eliminating the key projection matrices and sharing the output matrices $\mathbf{W}_o$, leading to fewer FLOPs compared to standard linear-attention-based Transformers. FixedVAR, despite containing more parameters due to a fixed weight per lag, avoids the computational overhead associated with dynamically generated weights, resulting in lower FLOPs than standard linear-attention Transformers.

## A.8. Theoretical Analysis of Robust Path Pruning: Bounded Dot-Product via RMSNorm

**Proposition.** Applying RMSNorm to query and value vectors bounds the magnitudes of the dot-products in temporal influence paths, thus preventing numerical instability in SAMoVAR.

**Proof.** We first recall that the query $\mathbf{q}_t^{(l)}$ and value vectors $\mathbf{v}_i^{(l)}$ are normalized by RMSNorm as:

$$\mathbf{q}_t^{(l)} = \text{RMSNorm}(\mathbf{x}_t^{(1)}\mathbf{W}_q^{(l)}), \quad \mathbf{v}_i^{(l)} = \text{RMSNorm}(\mathbf{x}_i^{(1)}\mathbf{W}_v^{(l)}),$$

where RMSNorm is defined as:

$$\text{RMSNorm}(\mathbf{y}) = \frac{\mathbf{y}}{\sqrt{\frac{1}{d}\sum_{j=1}^d y_j^2}} \odot \mathbf{g}.$$

Then, the absolute value of their dot-product can be bounded as follows:

$$|\mathbf{v}_i^{(l)\top}\mathbf{q}_t^{(l)}| = \left| \sum_{j=1}^d v_{i,j}^{(l)} q_{t,j}^{(l)} \right| \leq \|\mathbf{v}_i^{(l)}\|_2 \|\mathbf{q}_t^{(l)}\|_2 \approx \|\mathbf{g}_v^{(l)}\|_2 \|\mathbf{g}_q^{(l)}\|_2.$$

Given that the gain parameters $\mathbf{g}$ are typically initialized with small values, e.g., $\mathbf{g} \sim \mathcal{N}(0, 1/d)$, the resulting upper bound is approximately 1. Thus, for a given temporal influence path spanning $l$ layers:

$$|\mathbf{P}_{t,j,i_1,\ldots,i_{l-1}}^{(l)}| = |\mathbf{v}_{i_1}^{(l)\top}\mathbf{q}_t^{(l)}| \cdot |\mathbf{v}_{i_2}^{(l-1)\top}\mathbf{q}_{i_1}^{(l-1)}| \cdots |\mathbf{v}_j^{(1)\top}\mathbf{q}_{i_{l-1}}^{(1)}| \leq 1.$$

In practice, these dot-products are smaller than one, ensuring numerical stability and preventing gradient explosion during training.

## A.9. Theoretical Analysis of Robust Path Pruning: Orthogonality Probability in High Dimensions

**Proposition.** As the dimension $d$ increases, randomly initialized vectors become increasingly orthogonal with high probability, naturally pruning irrelevant temporal influence paths in SAMoVAR.

**Proof.** Consider two normalized random vectors $\hat{\mathbf{q}}, \hat{\mathbf{v}} \in \mathbb{R}^d$, each component initialized independently from a symmetric distribution with mean zero and finite variance. The dot-product of these vectors is given by:

$$\hat{\mathbf{q}}^\top \hat{\mathbf{v}} = \sum_{i=1}^d \hat{q}_i \hat{v}_i.$$

By the central limit theorem, as $d \to \infty$, the distribution of the dot-product converges to a Gaussian distribution:

$$\hat{\mathbf{q}}^\top \hat{\mathbf{v}} \xrightarrow{d \to \infty} \mathcal{N}\left(0, \frac{1}{d}\right).$$

Given a small threshold $\epsilon > 0$, we can compute the probability that the absolute value of their dot-product is below $\epsilon$:

$$P(|\hat{\mathbf{q}}^\top \hat{\mathbf{v}}| < \epsilon) \approx 2\Phi\left(\epsilon\sqrt{d}\right) - 1,$$

where $\Phi$ denotes the cumulative distribution function of the standard Gaussian distribution.

As dimension $d$ increases, this probability approaches 1, implying that with high probability, random vectors become nearly orthogonal. For temporal influence paths defined as:

$$\mathbf{P}^{(l)}_{t,j,i_1,\ldots,i_{l-1}} = (\mathbf{v}^{(l)\top}_{i_1}\mathbf{q}^{(l)}_t)(\mathbf{v}^{(l-1)\top}_{i_2}\mathbf{q}^{(l-1)}_{i_1})\cdots(\mathbf{v}^{(1)\top}_{j}\mathbf{q}^{(1)}_{i_{l-1}}),$$

the probability of any path having small or near-zero magnitude grows with dimension, thus naturally "pruning" irrelevant paths. During training, important paths are enhanced by gradient-based updates, selectively preserving informative temporal influence patterns.

