# OpenReview forum: "Linear Transformers as VAR Models:  Aligning Autoregressive Attention Mechanisms with Autoregressive Forecasting"
_ICML.cc/2025/Conference — ICML 2025 poster_

### Official Review · Reviewer_NbcJ · 2025-02-28

**Overall Recommendation:** 3

**Summary:**

The paper addresses time series forecasting by aligning linear-attention Transformer with vector autoregressive (VAR). The authors reveal that while single-layer linear attention mechanisms naturally exhibit a dynamic VAR structure, but multi-layer Transformers can misalign with the autoregressive forecasting objective. To address this, they propose a linear-attention Transformer variant that incorporates dynamic VAR weights by reorganizing input-output flows. Empirical results show the proposed models improve the performance and interpretability compared to several baseline models.

**Claims And Evidence:**

The claims made in the paper are generally supported, for example, the authors provide a theoretical analysis of how linear attention can be interpreted as a VAR model. However, some claims are made based on empirical results from previous works. They claim that "linear attention can outperform vanilla attention in time series forecasting tasks". This may not be the consensus in the field. It is also inconsistent with the evaluations (Table 1), where linear-attention Transformer can not generally outperform the vanilla ones. Some claims may be oversimplified: "Replacing $\sigma$ with an identity simplifies it to linear attention," Which linear attention is referred to? This assumption makes the Attn() a linear transformation of X.  Some claims can be biased: "Transformers are effective for modeling complex sequence relationships (e.g., in NLP), their architecture conflicts with VAR’s goal of explicitly representing lag-based dependencies." The author does not provide evidence that explicit VAR can help in terms of training models for time series rather than natural language. Therefore, it is better to explain the "capability of modeling complex sequence relationships" and why the alignment with explicit VAR is important.

**Essential References Not Discussed:**

The paper can benefit from discussing more recent advancements in autoregressive Transformers for time series forecasting, such as TimesFM [ICML 2024]. Whether the method proposed in this paper can improve the performance on similar models?

**Experimental Designs Or Analyses:**

* I recommend the author to evaluate more baseline models, including more counterpart linear-attention Transformers (currently 2), and decoder-only Transformers (currently 0).
* How about the performance when using more Transformer layers? Is the training process of SAMoVAR stable?

**Methods And Evaluation Criteria:**

The authors use standard benchmark datasets for time series forecasting, such as Weather, Solar, and ETT, which are widely adopted in the previous works.

**Other Comments Or Suggestions:**

See Above.

**Other Strengths And Weaknesses:**

* Strength: The paper presents a theoretically proof to reveal the similarity between simplified linear Transformers with VAR models.
* Weakness: The motivations of aligning linear Transformer with VAR are not well supported. To my best knowledge, linear Transformers are not the dominant choice in time series forecasting.

**Questions For Authors:**

Is there information leakage when the above method is used for training encoder-only models? Although the model is autoregressive supervised by the groundtruth in multi-step prediction, there is no groundtruth at each step during inference.

**Relation To Broader Scientific Literature:**

This paper focuses on refining Transformers for time series forecasting.

**Theoretical Claims:**

The proof in Section 3 appears to be correct, but there are some incorrect assumptions: (1) Section 2.2: Equation (1) omits the scaling factor, (2) The recurrent form of Attention relies on the causal mask. However, I am not certain about the proposed architecture adopting the causal mask. It seems that Transformers in Table 1 are not decoder-only ones? Please correct me if I'm mistaken..

---

> ### Author Rebuttal · Authors · 2025-04-01
>
> >Linear attention outperform ... may not be the consensus
> >Linear Transformers are not the dominant ...
>
> Thank you for the question. Our claim is based on [1] (https://arxiv.org/abs/2410.03159, cited at line 479), which suggests AR linear attention may outperform vanilla attention in TSF. We also support this by referencing linear attention's performance in other position-sensitive domains.
>
> We will revise the text to include qualifiers (e.g., sometimes) for clarity. Additionally, we provide more results at [Link (Table 1)](https://anonymous.4open.science/r/SAMoVAR-Rebuttal-F83C) comparing linear and softmax attention, showing advantage.
>
> Prior works (DLinear, PatchTST ...) have noted overfitting in TSF Transformers. Softmax attention’s strong expressiveness can overfit noise. In contrast, linear attention, due to the linearity of query, key, and value, retains an AR form [1] and also introduces a dynamic VAR structure:
>
> AR form of attention [1]:
>
> $$
> \mathbf{o}\_t = \sum\_{i=1}^t \mathbf{w}\_{t,i} \mathbf{v}\_i \ , \mathbf{w}\_{t,i} = \mathbf{q}\_t \mathbf{k}\_i^\top \in \mathbb{R}^d
> $$
>
> VAR form of linear attention [This paper]:
>
> $$
> \mathbf{o}\_t = \sum_{i=1}^t \mathbf{k}\_i \mathbf{A}\_{t,i} \ , \mathbf{A}\_{t,i} = \mathbf{q}\_t^\top \mathbf{v}\_i  \in \mathbb{R}^{d \times d}
> $$
>
> This allows for dual linear interpretability, with stronger regularization, better suited for channel- and token-wise linear mixing at each layer, suited for TSF data commonly modeled with AR/VAR structures.
>
> >Inconsistent with (Table 1) ...
>
> This aligns with Table 1: LinTrans outperforms softmax-based models (iTransformer, PatchTST, EncFormer). The only exception is CATS, which is MLP-based and does not use softmax attention.
>
> >Some claims may be oversimplified: Replacing $\sigma$ ...
>
> Apologies—this should read “simplified to linear attention with a linear kernel.” This is a standard baseline in recent works like Gated Linear Attention and RetNet. We’ll clarify this in the revision.
>
> >Evidence that explicit VAR ... "capability of modeling complex ..."
>
> Due to space limits, please refer to our response to Reviewer RLuA for discussion on NLP data.
>
> > Incorrect assumptions: (1) ..., (2) ...
>
> (1) Apologies—we’ll clarify this. The scaling factor acts like a softmax temperature and doesn’t affect analysis.
>
> (2) Yes, all attention structures in our methodology use decoder-only attention with a causal mask. A comparison is provided below:
>
> Encoder-only (without causal mask):
>
> $$
> \mathbf{O} = \mathbf{Q} \mathbf{K}^\top \mathbf{V} \\
> \mathbf{o}\_t = \mathbf{q}\_t \sum_{i=1}^N \mathbf{k}\_i^\top \mathbf{v}\_i
> $$
>
> Decoder-only (with causal mask, used in this paper):
>
> $$
> \mathbf{O} = \mathbf{M} \odot (\mathbf{Q} \mathbf{K}^\top) \mathbf{V} \\
> \mathbf{o}\_t = \mathbf{q}\_t \sum\_{i=1}^t \mathbf{k}\_i^\top \mathbf{v}\_i
> $$
>
> Timesteps are $\{1, ..., t, ..., N\}$. For encoder-only, $\mathbf{q}_t$ sees all tokens; for decoder-only, it sees only up to step $t$. This notation is standard in linear attention literature.
>
> >Evaluate more baseline
>
> Note that LinTrans uses decoder-only attention, while iTransformer, PatchTST, and EncFormer use encoder-only softmax attention. Additional results at [Link (Table 1)](https://anonymous.4open.science/r/SAMoVAR-Rebuttal-F83C) compare with AR softmax and gated linear attention.
>
> >Performance ... more layers
>
> Table 2 shows that ~3 layers of SAMoVAR work well across most datasets, with other choices showing no major drop in performance. The $l$-th layer output aggregates results from $n^{\text{path}}_{t,j,l}$ paths per key observation:
>
> $$
> \mathbf{o}\_t^{(l)} = \sum\_{\text{All Paths}} \mathbf{P}\_{t,j,\{i\_1, \cdots, i\_{l-1}\}}^{(l)} \mathbf{k}\_i^{(1)\top} = \sum\_{\text{All Paths}} \mathbf{v}\_{i\_1}^{(l)\top} [\mathbf{q}\_t^{(l)} \,\mathbf{v}\_{i\_2}^{(l-1)\top} \mathbf{q}\_{i\_1}^{(l-1)} \;\cdots\; \mathbf{v}\_{j}^{(1)\top}] \mathbf{q}\_{i_{l-1}}^{(1)} \mathbf{k}\_i^{(1)\top}
> $$
>
> Each path is scaled by the bracketed scalar. Normalizing q and k keeps inner products close to <1, avoiding instability. The only risk of deeper models comes from the increase in path count ($n^{\text{path}}_{t,j,l}$), which may cause overfitting of too many dynamics for longer lags.
>
> >More recent advancements ... TimesFM ...
>
> We will add a Related Works section for recent TSF advances to support the use of AR Lin Attn in TSF.
>
> >Is there information leakage...
>
> Our AR/ARX tokenization adds an AR loss between non-overlapping PatchTST tokens. Each token covers the full horizon $L_P$, allowing us to follow the same training/testing setup as prior works, with only one added loss term. The decoder-only enables effective training, functionally similar to shorter-context data augmentation or varying-context multitask training.
>
> This allows us to integrate AR attention without changing the experimental setup. Like previous works, we use one-step prediction covering $L_P$, avoiding iterative prediction and any risk of data leakage.

---

> > ### Comment · Reviewer_NbcJ · 2025-04-08
> >
> > Thanks for the detailed rebuttal. I have raised the score as the authors have resolved most of the concerns.

---

### Official Review · Reviewer_pxAU · 2025-03-05

**Overall Recommendation:** 3

**Summary:**

This paper addresses the misalignment between deep Transformer architectures and autoregressive objectives in time series forecasting (TSF), proposing Structural Aligned Mixture of VAR (SAMoVAR) to integrate interpretable dynamic VAR weights into multi-layer linear Transformers. By reorganizing the input-output flow and aligning the structure with VAR models, SAMoVAR enhances the ability to capture data generative processes, achieving improved performance, interpretability, and efficiency compared to state-of-the-art TSF models through experiments on synthetic and real-world datasets.

## update after rebuttal
I have reviewed your responses and those to other reviewers. While some concerns have been addressed, I remain unconvinced regarding the theoretical insights or experimental results on capturing periodicity and non-linear patterns. Therefore, I will maintain my original score.

**Claims And Evidence:**

The assertion that single-layer linear attention aligns with a dynamic VAR structure is backed by rigorous theoretical derivations (Equations 2–3) and validated through synthetic experiments (Figures 3–4). Additionally, the improved performance of SAMoVAR over baseline models is demonstrated across multiple datasets, with significant reductions in MSE (e.g., 30% on the Solar dataset compared to previous models), providing strong empirical support for the effectiveness of the proposed model.

However, the claim that SAMoVAR provides interpretable dynamic VAR weights could be further strengthened. While visualizations (e.g., Figure 5) illustrate the temporal influence paths, they lack quantitative analysis.

**Essential References Not Discussed:**

Limited discussion of concurrent work on efficient attention mechanisms (e.g., Retentive Networks [Sun et al. 2023], H3 [Dao et al. 2023]), which also achieve linear complexity.

**Experimental Designs Or Analyses:**

Batch size adjustments for large datasets (e.g., PEMS07 excluded due to batch size=1) may introduce bias. No sensitivity analysis is provided.

**Methods And Evaluation Criteria:**

The paper introduces "Robust Path Pruning" to ensure numerical stability and control weight variance in SAMoVAR. While the ablation studies show that removing RMSNorm degrades performance, further validation is needed to fully support the claims around path pruning. Specifically, the paper lacks detailed metrics on the percentage of paths pruned and the correlation between pruned paths and model performance. Additionally, theoretical analysis on the probability of query-value orthogonality and comparison to active pruning methods would strengthen the claims. Assessing the impact of pruning on the model’s ability to capture long-term dependencies and measuring the rank of the resulting weight matrices would also provide deeper insights into the effectiveness of the pruning mechanism.

**Other Comments Or Suggestions:**

The current model framework diagram is insufficient for fully understanding the proposed mechanisms, especially the Temporal Influence Path and Robust Path Pruning. Enhancing the visual representation with more detailed diagrams or flowcharts would significantly aid in clarifying these complex components and their integration into the overall architecture.

**Other Strengths And Weaknesses:**

Strengths:

- Originality: The integration of VAR theory into linear attention is novel, offering a fresh perspective on Transformer interpretability.

- Significance: The method’s efficiency and performance improvements are practically valuable for real-world TSF applications.

Weaknesses:

- Clarity: The temporal influence path concept (Sec. 4.1) is overly abstract. A step-by-step example (e.g., for 2-layer SAMoVAR) would improve readability.

**Questions For Authors:**

1.Time series often exhibit periodic and non-linear patterns, especially in long-term forecasting. How does SAMoVAR effectively capture these characteristics, and are there theoretical insights or experiments demonstrating its ability to handle periodicity and non-linear interactions?

2.Irregular time series data are common in healthcare applications. Can the authors discuss SAMoVAR’s potential advantages in handling such data, particularly in medical forecasting tasks?

3.The paper claims that SAMoVAR maintains O(N) complexity, but the new architecture might introduce additional overhead. Can the authors provide a detailed complexity analysis, especially for the Temporal Influence Path mechanism?

4.The experiments show SAMoVAR’s effectiveness with specific hyperparameters. How robust are these settings across different datasets, and can the authors provide guidelines for selecting optimal configurations in various scenarios?

**Relation To Broader Scientific Literature:**

The work bridges classical VAR models and modern linear Transformers, extending prior efforts like [Katharopoulos et al. 2020] (linear attention as RNNs) and [Zeng et al. 2023] (Transformer-VAR comparisons).

**Theoretical Claims:**

Multi-Layer Alignment: The claim that "multi-layer linear attention aligns with VAR" (Sec. 4.1) relies on recursive expansions (Eq. 5) but lacks formal proof. For instance, it is unclear whether the cumulative paths preserve VAR properties (e.g., stationarity).

Stability Guarantees: No theoretical analysis ensures that path pruning or RMSNorm prevents gradient explosion in deep layers.

---

> ### Author Rebuttal · Authors · 2025-04-01
>
> >No theoretical analysis for robust path pruning
>
> Thank you for the insightful question. We provide a proof sketch below.
>
> **Theorem 1**: RMSNorm applied to query and value vectors bounds the magnitudes of dot products in temporal influence paths, preventing numerical instability.
>
> **Proof Sketch**:
> Query and value vectors are normalized using RMSNorm:
>
> $$\mathbf{q}_t^{(l)} = \text{RMSNorm}(\mathbf{x}_t^{(1)}\mathbf{W}_q^{(l)}), \quad \mathbf{v}_i^{(l)} = \text{RMSNorm}(\mathbf{x}_i^{(1)}\mathbf{W}_v^{(l)})$$
>
> where:
>
> $$\text{RMSNorm}(\mathbf{y}) = \frac{\mathbf{y}}{\sqrt{\frac{1}{d}\sum_{j=1}^{d} y_j^2}} \odot \mathbf{g}$$
>
> This bounds their dot product:
>
> $$|\mathbf{v}\_{i}^{(l)\top} \mathbf{q}\_{t}^{(l)}| \leq \|\mathbf{g}\_v^{(l)}\|_2 \cdot \|\mathbf{g}\_q^{(l)}\|_2 \approx 1$$
>
> with proper initialization of small gain parameters $\mathbf{g}$ (e.g., $\sim \mathcal{N}(0,\frac{1}{d})$).
>
> For a temporal influence path with $l$ layers:
>
> $$|\mathbf{P}\_{t,j,\{i_1,...,i\_{l-1}\}}^{(l)}| = |\mathbf{v}\_{i_1}^{(l)\top} \mathbf{q}\_t^{(l)}| \cdot ... \cdot |\mathbf{v}\_{j}^{(1)\top} \mathbf{q}\_{i\_{l-1}}^{(1)}| \leq 1$$
>
> In practice, path magnitudes decrease with depth as dot products are typically smaller than 1.
>
> **Theorem 2**: As dimension $d$ increases, random vectors become increasingly orthogonal, naturally pruning unnecessary paths.
>
> **Proof Sketch**:
> For normalized vectors $\hat{\mathbf{q}}$ and $\hat{\mathbf{v}}$ with random initialization, their dot product:
>
> $$\hat{\mathbf{q}}^T \hat{\mathbf{v}} = \sum_{i=1}^d \hat{q}\_i \hat{v}\_i$$
>
> has mean 0 and variance $\frac{1}{d}$. By the Central Limit Theorem, for large $d$, this approaches $\mathcal{N}(0,\frac{1}{d})$.
>
> For threshold $\epsilon > 0$, the probability of near-orthogonality is:
>
> $$P(|\hat{\mathbf{q}}^T \hat{\mathbf{v}}| < \epsilon) \approx 2\Phi\left(\epsilon\sqrt{d}\right) - 1$$
>
> which approaches 1 as $d$ increases.
>
> For a path:
>
> $$\mathbf{P}\_{t,j,\{i_1,...,i\_{l-1}\}}^{(l)} = \mathbf{v}\_{i\_1}^{(l)\top} \mathbf{q}\_t^{(l)} \cdot ... \cdot \mathbf{v}\_{j}^{(1)\top} \mathbf{q}\_{i_{l-1}}^{(1)}$$
>
> With larger $d$, the likelihood of a near-zero dot product increases, pruning irrelevant paths while training strengthens important ones.
>
> >The claim that SAMoVAR provides interpretable ... could be further strengthened.
>
> Our current temporal influence path visualization offers interpretability similar to attention maps but does not yet match that of traditional MLE-based VAR models. This paper focuses on SAMoVAR’s predictive performance, though full statistical interpretability is possible through:
>
> 1. Initializing query/value weights via SVD of training set covariance.
> 2. Breaking symmetry by fixing an element in the q/v weight matrices.
> 3. Removing MLPs, pre-normalization, patching, and projections for full linearity.
> 4. Training with MLE loss.
> 5. Estimating parameter uncertainty using asymptotic covariance for standard errors.
> 6. Applying the Delta method to propagate uncertainty in dynamic VAR weights.
>
> These changes would enable full interpretability but reduce model flexibility and accuracy. We’re happy to share our ongoing works in this direction, though it is beyond this paper’s current scope.
>
> >Whether the cumulative paths preserve VAR properties (e.g., stationarity)
> >Proof of VAR structure
>
> SAMoVAR is not constrained to classic stationarity. Enforcing eigenvalues within the unit circle, especially with long lags, would limit expressiveness and harm predictive power. As for the structural derivation of the model, Please see the proof in our response to Reviewer aK3c. Thank you!
>
> >Batch size adjustments may introduce bias.
>
> As noted in Appendix (line 707), we use gradient accumulation to maintain an effective batch size of 32 when reducing batch size. A reference will be added in the revision.
>
> >Ground-truth VAR parameters (Sec. 5.1) are not described, limiting reproducibility.
>
> The VAR parameters are shown in the top-left heatmap of Figure 3. All experiments and code from Section 5.1 are available in `visualization/model_visualize.ipynb` to ensure maximized reproducibility.
>
> >non-linear patterns ... ability to handle periodicity and non-linear interactions...
>
> Due to space limits, please refer to our response to **Reviewer RLuA** for discussion on NLP and complex data.
>
> >SAMoVAR... in medical forecasting tasks
>
> We use standard TSF benchmarks, which contain irregular patterns in some data (like Weather, ETTh1, ETTh2). SAMoVAR performs well, suggesting potential for irregular medical data.
>
> >Proof of SAMoVAR maintains O(N) complexity
>
> Due to space limits, please see the **proof** in our response to **Reviewer ZzwP**.
>
> >guidelines for selecting optimal configurations
>
> We use fixed hyperparameters across all datasets. The fixed $L_I$ settings highlight robustness compared to baselines needing $L_I$ tuning. See our response to Reviewer ZzwP, and Appendix A.3, where full configuration are ready to use without any modification.

---

### Official Review · Reviewer_aK3c · 2025-03-13

**Overall Recommendation:** 3

**Summary:**

This paper demonstrates that a single linear attention layer behaves like a dynamic VAR model and that deeper Transformers can be restructured to align with autoregressive objectives. Based on these insights, the authors introduce SAMoVAR, a Transformer variant that leverages dynamic VAR weights to enhance forecasting performance, interpretability, and efficiency.

Update after rebuttal: Thanks for the detailed rebuttal. I appreciate the authors' efforts in addressing my concerns and helping me better understand the work. Therefore, I will maintain my original rating.

**Claims And Evidence:**

Yes.

**Essential References Not Discussed:**

No.

**Experimental Designs Or Analyses:**

Yes.

**Methods And Evaluation Criteria:**

Yes.

**Other Comments Or Suggestions:**

See strengths and weaknesses.

**Other Strengths And Weaknesses:**

I am not very familiar with the time series forecasting task, but the paper reads logically and intuitively.

Strengths:
1. The motivation behind the proposal of SAMoVAR is well-founded.
2. The theoretical explanations are comprehensive and clear.
3. The experiments and the analysis of experimental results are thorough and convincing.

Question:

Why does the validation loss in Figure 4 exhibit significant oscillations during the training process?

Minor Issue:

Line 18: The paper refers to VAR before defining it.

**Questions For Authors:**

See strengths and weaknesses.

**Relation To Broader Scientific Literature:**

The authors introduce SAMoVAR, a Transformer variant that leverages dynamic VAR weights to enhance forecasting performance, interpretability, and efficiency.

**Theoretical Claims:**

N/A.

---

> ### Author Rebuttal · Authors · 2025-04-01
>
> >Why does the validation loss in Figure 4 exhibit significant oscillations during the training process?
>
> Thank you very much for taking the time to read our paper and for acknowledging its contributions.
>
> We apologize for the misunderstanding. In Figure 4, we cropped the Y-axis of the validation loss curve. Like the training loss, it starts around 0.1 and decreases over epochs. However, using 0.1 as the upper bound would make the convergence of all three curves indistinguishable in this compact figure. As shown, the validation loss fluctuates within ±0.002 in later epochs, consistent with the training loss scale.
>
> Notably, SAMoVAR surpasses the final performance of the other two baselines after just 30 epochs. The Y-axis scaling was chosen purely to highlight this advantage for clarity.
>
> You can verify more of this in our code repository: see `visualization/model_visualize.ipynb`, specifically `In [5]` and `Out [5]`, where we use `plt.ylim(-0.01, 0.1)` for the training loss and `plt.ylim(0.0025, 0.0085)` for the validation loss.
>
> >Line 18: The paper refers to VAR before defining it.
>
> Thank you for pointing this out. We will ensure that VAR is properly defined upon its first appearance in the revision.
>
> >Theoretical Claims:
> >N/A.
>
> Thank you for your positive feedback. Even though you didn’t explicitly request it, we’re happy to provide a more detailed derivation of how SAMoVAR’s multi-layer linear attention can be expressed as a VAR structure, to further support your understanding. If you're already familiar with this, feel free to skip the following explanation. Thank you again!
>
>
> **Proposition** For any $l \geq 1$, the output $\mathbf{o}_t^{(l)}$ can be expressed as VAR structure:
> $$\mathbf{o}\_t^{(l)\top} = \sum\_{j=1}^t \mathbf{B}\_{t,j}^{(l)} \mathbf{x}\_j^{(1)\top}$$
>
> **Proof by Induction:**
>
> **Base Case (l=1)** For a single linear attention layer, we have:
> $$\mathbf{o}\_t^{(1)\top} = \sum_{i=1}^t \mathbf{A}\_{t,i}^{(1)} \mathbf{k}\_i^{(1)\top} = \sum\_{i=1}^t \mathbf{A}\_{t,i}^{(1)} \mathbf{x}\_i^{(1)\top}$$
>
> Since $\mathbf{k}\_i^{(1)} = \mathbf{x}\_i^{(1)}$, we define $\mathbf{B}\_{t,i}^{(1)} = \mathbf{A}\_{t,i}^{(1)}$, which gives us the VAR form for the first layer.
>
> **Inductive Step** The key insight is the relationship between outputs and inputs across layers. When we use $\mathbf{k}\_i^{(l+1)} = \mathbf{o}\_i^{(l)}$, we get:
> $$\mathbf{o}\_t^{(l+1)\top} = \sum\_{i=1}^t \mathbf{A}\_{t,i}^{(l+1)} \mathbf{o}\_i^{(l)\top}$$
>
> Substituting the inductive hypothesis and rearranging the sums:
> $$\begin{aligned}
> \mathbf{o}\_t^{(l+1)\top} &= \sum\_{i=1}^t \mathbf{A}\_{t,i}^{(l+1)} \left(\sum\_{j=1}^i \mathbf{B}\_{i,j}^{(l)} \mathbf{x}\_j^{(1)\top}\right) \\
> &= \sum\_{j=1}^t \sum\_{i=j}^t \mathbf{A}\_{t,i}^{(l+1)} \mathbf{B}\_{i,j}^{(l)} \mathbf{x}\_j^{(1)\top} \\
> &= \sum\_{j=1}^t \left(\sum_{i=j}^t \mathbf{A}\_{t,i}^{(l+1)} \mathbf{B}\_{i,j}^{(l)}\right) \mathbf{x}\_j^{(1)\top}
> \end{aligned}$$
>
> Therefore:
> $$\mathbf{o}\_t^{(l+1)\top} = \sum\_{j=1}^t \mathbf{B}\_{t,j}^{(l+1)} \mathbf{x}\_j^{(1)\top}$$
>
> Where:
> $$\mathbf{B}\_{t,j}^{(l+1)} = \sum\_{i=j}^t \mathbf{A}\_{t,i}^{(l+1)} \mathbf{B}\_{i,j}^{(l)}$$
>
> This recursive definition of $\mathbf{B}_{t,j}^{(l)}$ captures all possible influence paths from time $j$ to time $t$ through $l-1$ intermediate points, confirming that multi-layer linear attention maintains an VAR structure throughout the network.

---

### Official Review · Reviewer_RLuA · 2025-03-18

**Overall Recommendation:** 3

**Summary:**

This paper demonstrates that autoregressive linear attention can be interpreted as a rank-1 vector autoregressive (VAR) model. Building upon this perspective, the authors introduce SAMoVAR, a novel model achieved by stacking multiple linear attention layers. SAMoVAR overcomes the inherent rank-1 limitation of linear attention, thus enhancing the model's expressiveness and forecasting performance on time series tasks.

**Claims And Evidence:**

The claims presented in the paper are supported by both theoretical analyses and empirical evidence.

**Essential References Not Discussed:**

The references are sufficiently comprehensive. However, integrating more explicit discussions about linear attention approaches prevalent in the broader literature (e.g., https://arxiv.org/pdf/2210.10340; https://arxiv.org/pdf/2006.16236; https://arxiv.org/pdf/2405.13956) would strengthen connections to related work.

**Experimental Designs Or Analyses:**

The main comparative experiments are well-designed. However, the explainability section lacks clarity. This aspect needs further clarification (see Questions for Authors).

**Methods And Evaluation Criteria:**

The authors evaluate the proposed method across multiple multivariate time series forecasting datasets, demonstrating consistent improvements over state-of-the-art methods on most datasets.

**Other Comments Or Suggestions:**

Formatting / Typos:

1. The page header is still shown as "Submission and Formatting Instructions for ICML 2025".

**Other Strengths And Weaknesses:**

N/A

**Questions For Authors:**

[Q1] Could you provide insight into how your VAR framework addresses or mitigates common issues with linear attention and its variants, such as attention dilution and unbounded gradients described by Qin et al. (https://arxiv.org/pdf/2210.10340)?

Under the VAR interpretation, what specifically causes these issues in linear attention and its variants?

Which of these known problems does SAMoVAR solve or alleviate, and which problems persist? Please clarify the underlying reasons.

[Q2] The current explainability analyses primarily highlight smoother temporal weights and locality. Could you further clarify how SAMoVAR's VAR structure explicitly enhances interpretability regarding task-specific insights?

[Q3] Have you evaluated SAMoVAR in language modeling tasks? If yes, how does its performance compare with traditional linear attention methods?

**Relation To Broader Scientific Literature:**

The paper provides a novel perspective by framing linear attention in the VAR context, potentially bridging VAR models and linear attention architectures. However, connections to the linear attention literature could be strengthened further, particularly by directly comparing SAMoVAR with Linear Attention and its variants. Such comparisons could enhance accessibility for audiences from the Linear Transformer and LLM communities.

**Theoretical Claims:**

The theoretical derivations in the paper are clearly presented. I did not identify any major issues in the theoretical analysis.

---

> ### Author Rebuttal · Authors · 2025-04-01
>
> >However, integrating more explicit discussions about linear attention...
>
> Thank you for the valuable suggestion. Currently, related literature is only discussed in the Introduction and Background. We will add a Related Work section in the revision for a more thorough and structured discussion.
>
> >The page header is still shown as "Submission and Formatting Instructions for ICML 2025".
>
> Thank you for pointing this out. We will ensure the correct header format is used in the revision to comply with the guidelines.
>
> >Could you provide insight into how your VAR framework addresses or mitigates common issues with linear attention and its variants
>
> Thank you for the insightful question. Prior work addresses the unbounded gradient issue by normalizing $ \mathbf{o}_t $. Similarly, in SAMoVAR, we apply RMSNorm to stabilize computations:
> $$
> \phi(\mathbf{q}_t)^{(l)} = \text{RMSNorm}(\mathbf{x}_t^{(1)} \mathbf{W}_q^{(l)}),
> \quad \phi(\mathbf{k}_i)^{(1)} = \mathbf{x}_i^{(1)},
> \quad \phi(\mathbf{k}_i)^{(l)} = \mathbf{o}_i^{(l-1)},
> \quad \mathbf{v}_i^{(l)} = \text{RMSNorm}(\mathbf{x}_i^{(1)} \mathbf{W}_v^{(l)})
> $$
> Note that $ \mathbf{x}_i^{(1)} $ is also normalized beforehand. This aligns well with the approach used in the referenced work.
>
> The notion of *attention dilution* suggests that softmax attention tends to focus on local token relationships. However, we caution against a potential causality issue here: in NLP, softmax attention appears to model local relationships not because of an inherent bias, but because NLP data often demands such modeling. Thanks to its strong selectivity, softmax attention can naturally focus on local context even without explicit inductive bias: unlike linear attention, which tends to produce more diffuse attention patterns due to weaker selectivity. In contrast, for time series data, softmax attention, being a smooth approximation of the max operation, can overly concentrate on specific intervals. Yet time series typically exhibit stable temporal patterns such as lagged autocorrelations, cycles, seasonality, and trends. These effects are less variable than token-wise dependencies in NLP. Therefore, a VAR-based inductive bias is better suited to capture these temporal consistent structures, avoiding the overly sharp, localized focus induced by softmax attention, which may result in overfitting as stated in previous Transformer-based TSF works.
>
> >The current explainability analyses primarily highlight smoother temporal weights and locality. Could you further clarify how SAMoVAR's VAR structure explicitly enhances interpretability regarding task-specific insights?
>
> Thank you for the question. As noted earlier, attention in NLP rarely models effects tied to absolute lag positions, since the context relationships vary greatly across inputs. This differs significantly from the stable, linearizable lag patterns seen in autocorrelated VAR processes.
>
> In contrast, tasks involving time series, images, speech, and those in Long Range Arena (https://arxiv.org/abs/2011.04006) are more position-sensitive. Recent studies on efficient linear attention often evaluate on these domains. While many linear attention variants struggle to outperform vanilla Transformers on NLP tasks, they often surpass them on position-related tasks, especially LRA benchmarks.
>
> Our interpretation of linear attention as a form of VAR modeling helps explain this: VAR weights can explicitly capture stable positional patterns, such as autocorrelations in token space, offering a better inductive bias in these settings with consistent positional patterns.
>
> >Have you evaluated SAMoVAR in language modeling tasks? If yes, how does its performance compare with traditional linear attention methods?
>
> Thank you for the valuable question. Building on the previous responses, you may understand that our positioning of SAMoVAR is as follows: it is particularly suited for data with clear autoregressive structure, strong autocorrelations, and stable positional effects, as seen in TSF data.
>
> For NLP tasks, SAMoVAR’s dynamic weight matrix would need to vary more significantly, similar to softmax, to adapt to diverse input patterns. This remains an interesting direction for future exploration.

---

### Official Review · Reviewer_ZzwP · 2025-03-23

**Overall Recommendation:** 3

**Summary:**

This paper proposes structural modifications to linear Transformer architectures to better align them with the Vector Autoregressive (VAR) framework, which is widely used in time series forecasting. The authors show that while a single-layer linear attention module can naturally express a dynamic VAR structure, standard multi-layer Transformer designs introduce misalignments that hinder interpretability and forecasting performance. To address this, they introduce SAMoVAR (Structural Aligned Mixture of VAR), a novel Transformer variant that reorganizes the attention and MLP layers to preserve a coherent dynamic VAR interpretation across multiple layers. The paper presents both theoretical formulations and empirical evaluations, demonstrating that SAMoVAR achieves improved accuracy, interpretability, and computational efficiency on a range of standard multivariate time series forecasting benchmarks, outperforming existing state-of-the-art models.

**Claims And Evidence:**

The primary claim of the paper is that the proposed SAMoVAR architecture improves over state-of-the-art Transformer-based models for time series forecasting in terms of accuracy, interpretability, and computational efficiency. This claim is generally supported by empirical evidence: the authors report strong performance across a wide range of standard benchmarks, and the improvements in accuracy are consistent across datasets.

However, some concerns remain. First, while the reported results show SAMoVAR outperforming prior methods, the performance margins differ from those presented in original papers of some recent baselines, suggesting that the experimental setup may not fully match those prior works. Clarification on baseline implementations and training details would help validate the comparison.

Second, the claim of superior computational efficiency is less clearly supported. Table 8, which is meant to demonstrate SAMoVAR's efficiency, includes results that appear counterintuitive or internally inconsistent. These discrepancies warrant further clarification from the authors to substantiate the efficiency claim.

**Essential References Not Discussed:**

While not strictly required, given the popularity and rapid development in time series forecasting (TSF), it would be helpful for the authors to discuss very recent advances in the field. One such example is TimeMixer++: A General Time Series Pattern Machine for Universal Predictive Analysis by Wang et al., recently accepted at ICLR 2025. This work proposes a highly general and competitive TSF model, and a comparison—either empirical or conceptual—would help situate SAMoVAR more clearly within the current landscape of TSF methods.

**Experimental Designs Or Analyses:**

The experimental design and analysis in the paper are generally sound. The authors evaluate the proposed method across a diverse set of standard time series forecasting benchmarks and compare against strong, commonly used baselines. However, there are two notable concerns. First, the reported results for some prior state-of-the-art methods differ from those presented in their original papers, raising questions about reproducibility and consistency in implementation. Second, the compute efficiency claims—particularly those in Table 8—are not fully convincing and appear inconsistent or counterintuitive. These aspects would benefit from further clarification and verification.

**Methods And Evaluation Criteria:**

The proposed method and evaluation criteria are appropriate for the time series forecasting task. The paper follows standard practices in the field, employing widely-used multivariate TSF benchmark datasets and comparing against strong Transformer-based baselines. The experimental setup, including input/output lengths and tokenization strategies, is consistent with prior work, ensuring a fair and relevant evaluation of the proposed model.

**Other Comments Or Suggestions:**

No additional comments or suggestions.

**Other Strengths And Weaknesses:**

No additional strengths or weaknesses beyond those already discussed.

**Questions For Authors:**

1. Comparison to Recent SOTA Methods: Given the likely overlap between the authors of this paper and the recent CATs model, could the authors provide more discussion or empirical comparison to other recent strong TSF methods such as TimeMixer++ (ICLR 2025)? A clearer comparison would help better position SAMoVAR in the current TSF landscape.

2. Why SAMoVAR Works Better Than CATs: While the paper clearly explains how SAMoVAR is constructed, it is less clear why it performs better than prior models like CATs. Could the authors elaborate on the intuition or mechanisms that drive the performance gains?

3. Clarification of Table 8 (Compute Efficiency): The reported FLOPs in Table 8 appear to be inconsistent or counterintuitive for several models. Could the authors clarify how these values were computed and confirm their correctness? A more transparent explanation would improve the credibility of the compute efficiency claims.

**Relation To Broader Scientific Literature:**

This paper contributes to the growing body of work at the intersection of classical statistical models and deep learning architectures. By drawing a connection between Vector Autoregressive (VAR) structures and linear Transformer models, the authors offer a novel perspective on how autoregressive attention mechanisms can be restructured to align with traditional time series modeling principles. If the empirical results are verified, the proposed SAMoVAR model could represent a meaningful step forward in the design of interpretable and effective models for time series forecasting (TSF), potentially influencing future research in both the deep learning and econometrics communities.

**Theoretical Claims:**

The paper does not include formal theoretical claims or proofs. While it provides intuitive interpretations and derivations connecting linear attention mechanisms to dynamic VAR structures, these are presented in an informal manner without rigorous theoretical guarantees. This absence raises some concern regarding whether the observed performance gains stem from fundamental modeling improvements or from design choices that may implicitly favor the proposed method. Including formal analysis or clearer theoretical justification would strengthen the paper’s contribution.

---

> ### Author Rebuttal · Authors · 2025-04-01
>
> >Performance differ from those presented in original papers ...
>
> Thank you for your valuable question. For each baseline and output length $ L_P $, we run experiments with input lengths $ L_I \in \{512, 1024, 2048, 4096\} $ and report the best result. This avoids biases from models' sensitivity to input length, as noted in prior work. Our reported results may outperform those in the original papers, but never underperform. This evaluation strategy, also used in TiDE (https://arxiv.org/abs/2304.08424), ensures fairer and more strict comparison. Our problem setup follows DLinear, PatchTST, and iTransformer, and can be verified in our code repository (see `baseline.sh`).
>
> ```bash
> for pred_len in 96 192 336 720
> do
>
> for seq_len in 4096 2048 1024 512
> do
> ...
> ```
>
> >Computational efficiency is less clearly supported
> >Including formal analysis or clearer theoretical justification would strengthen the paper’s contribution.
>
> Thank you for the suggestion. Our writing style follows prior work of LinTrans (e.g., Katharopoulos et al., 2020; Yang et al., 2024), focusing more on structure derivations and insights. We’re happy to add rigorous proofs of computational efficiency in the revision. Please see below:
>
> **Proposition** SAMoVAR Attention has $ O(L) $ complexity with respect to sequence length $ L $.
>
> **Proof** We analyze Algorithm 1 (SAMoVAR) in the appendix. Let batch size be $ B $, model dimension $ D $, number of heads $ H $, per-head dimension $ d = D/H $, and number of attention layers $ L_{\text{attn}} $.
>
> **Preprocessing**:
> - **LU Matrix Generation**: Creating the invertible matrix $ \mathbf{D} $ via LU decomposition is independent of $ L $, with cost $ O(Hd^2) $ per layer.
>
> **Per-Layer Operations** (for each $ l = 1, ..., L_{\text{attn}} $):
>
> 1. **Linear Projections**: Computing $ Q^{(l)} $ and $ V^{(l)} $ costs $ O(LD^2) $.
>
> 2. **Cumulative State Update**: Recursively updating
>    $$
>    W_t = W_{t-1} + K_t \otimes V_t^{(l)}
>    $$
>    costs $ O(Hd^2) $ per step, totaling $ O(LHd^2) = O(LD^2/H) $.
>
> 3. **Output Computation**:
>    $$
>    Y_t = Q_t^{(l)} \otimes W_t
>    $$
>    also costs $ O(LD^2/H) $.
>
> **Structural Transformation**:
>    $$
>    Y_{t,\text{transformed}} = \text{einsum}('bhd,hde \rightarrow bhe', Y_t, \mathbf{D}^{-1})
>    $$
>
> has the same cost: $ O(LD^2/H) $.
>
> **Conclusion**: Each layer has total cost $ O(LD^2) $. Since the number of layers is constant, overall complexity is $ O(LD^2) $, linear in sequence length $ L $.
>
> >Comparison with TimeMixer++
>
> Thank you for the suggestion. We added a comparison with TimeMixer++ at [Link (Table 2)](https://anonymous.4open.science/r/SAMoVAR-Rebuttal-F83C), using its official implementation and hyperparameters from `TimeMixer_ETTh1_unify.sh`. We evaluated all $ L_I \in \{512, 1024, 2048, 4096\} $. Note that the $ L_I = 512 $ result here outperforms the original $ L_I = 96 $ result. Our findings show that TimeMixer++ have no robust advantage over prior baselines and performs worse at longer $L_I$, similar to patterns seen in the DLinear and PatchTST papers. Interestingly, while recent NLP models trend toward longer contexts, recent TSF models (e.g., iTransformer, TimeMixer++) keep $ L_I $ as 96, compared to 512 in PatchTST, which is counterintuitive given TSF data (on test set) is typically continuous and not limited the usage of long context. We hope our varying $ L_I $ setup encourages a shift for TSF community toward the standards in line with broader sequence modeling area.
>
> >Why SAMoVAR Better Than CATS
>
> Thank you for the insightful question. Our results show that LinTrans with ARX tokenization performs closely to CATS, suggesting that autoregressive loss (as short-context data augmentation) and varying-context multitask training allow linear attention to match MLP-based models.
>
> SAMoVAR goes further by explicitly modeling the VAR data generation process of MLP-constructed input sequences. You can view its MLP module as linearizing ARX-tokenized inputs, enabling dynamic VAR modeling via aligned linear attention. This offers a stronger inductive bias for TSF by better capturing lag, cycles in time series patterns compared to LinTrans.
>
> >Clarification of Table 8 (Efficiency)
>
> The computational costs in Table 8 align with our analysis. FLOPs and parameter counts are measured on ETTh1 with $ L_I = 512 $ for baselines and $ L_I = 1024 $ for linear attention models. All other hyperparameters follow the original ETTh1 settings.
>
> For linear attention variants, we set hidden dimensions as $ d = \text{int}(32 \sqrt{C}) $; with $ C=7, d = 64 $, this results in fewer parameters than PatchTST/iTransformer (128/256).
>
> SAMoVAR further reduces cost by removing all key projection matrices and sharing all output matrices $ \mathbf{W}_o $, leading to lower computational cost than LinTrans. FixedVAR, while having more parameters due to fixed weights per lag, avoids dynamic weight generation and thus has lower FLOPs than LinTrans.
>
> In summary, Table 8 fully aligns with our analysis.

---

### Decision · Program_Chairs · 2025-05-01

**Decision:**

Accept (poster)

**Comment:**

The paper first shows that single-layer linear attention module can naturally express a dynamic vector autoregressive model structure but standard multi-layer Transformer designs do not. They hypothesize that this mismatch, if resolved, could lead to a more performant class of neural networks for time series forecasting. To this end, they develop SAMoVAR (Structural Aligned Mixture of VAR), a novel Transformer variant that reorganizes the attention and MLP layers so that the computation across layers can be interpreted as a dynamic VAR model. The paper empirically tests the model on a standard suite of benchmarks in time series forecasting and shows that it achieves improved accuracy, interpretability, and computational efficiency. The discussions with the reviewers a] a comparison with TimeMixer (which was provided) b] an expansion on why SAMOVAR performs well c] a proof of how the multi-layer case of SAMOVAR can be seen as a VAR process and an analysis of robust path pruning. Overall, the above questions were responded to in the rebuttal and I encourage the authors to incorporate them into the revised manuscript. Given the novelty of the idea and the work's dive into understanding how to align time series based transformers with VAR models (as well as the latters in classical time-series forecasting problems) I lean towards acceptance.